# pH drives electron density fluctuations that enhance electric field-induced liquid flow

S. Pullanchery [1,6], S. Kulik[1,6], T. Schönfeldová[1], C. K. Egan[2], G. Cassone [3], A. Hassanali [2] ✉ & S. Roke [1,4,5] ✉

Liquid flow along a charged interface is commonly described by classical continuum theory, which represents the electric double layer by uniformly distributed point charges. The electrophoretic mobility of hydrophobic nanodroplets in water doubles in magnitude when the pH is varied from neutral to mildly basic (pH 7 → 11). Classical continuum theory predicts that this increase in mobility is due to an increased surface charge. Here, by combining all-optical measurements of surface charge and molecular structure, as well as electronic structure calculations, we show that surface charge and molecular structure at the nanodroplet surface are identical at neutral and mildly basic pH. We propose that the force that propels the droplets originates from two factors: Negative charge on the droplet surface due to charge transfer from and within water, and anisotropic gradients in the fluctuating polarization induced by the electric field. Both charge density fluctuations couple with the external electric field, and lead to droplet flow. Replacing chloride by hydroxide doubles both the charge conductivity via the Grotthuss mechanism, and the droplet mobility. This general mechanism deeply impacts a plethora of processes in biology, chemistry, and nanotechnology and provides an explanation of how pH influences hydrodynamic phenomena and the limitations of classical continuum theory currently used to rationalize these effects.

The liquid flow of an aqueous electrolyte solution in contact with a charged surface is commonly modeled using the incompressible Navier-Stokes equations that combine the continuity equation for the conservation of mass, with momentum conservation, and the Poisson-Boltzmann equation to describe the charge distribution[1]. This classical continuum approach works well to understand liquid flow in many situations, even down to the nanoscale[2–5]. However, this description relies on assumptions that are ultimately not valid: A structureless dielectric continuum, strong water-surface interactions to avoid liquid-surface slip, ions/charged groups as the only charge carriers, represented by point charges and giving rise to uniform charge distributions[1]. Electronic polarization is also not explicitly included within this context. Therefore, theoretical considerations aimed at investigating the use of a continuum vs atomistic description[6,7], the inclusion of dipolar (and possibly higher multipolar) interface contributions of oriented water[8], surface friction and slip[9–13], the inclusion of explicit distribution of ions in the electric double layer[14], and more recently, even the importance of electronic excitations on the flow[13] have been considered. These works describe how diverse effects impact electrophoretic phenomena and more generally, nanofluidics.

However, determining the correct representation of liquid flow along charged surfaces requires explicitly verifying the various

---

[1]Laboratory for fundamental BioPhotonics, Institute of Bioengineering (IBI), School of Engineering (STI), École Polytechnique Fédérale de Lausanne (EPFL), Lausanne, Switzerland. [2]International Centre for Theoretical Physics, Trieste, Italy. [3]Institute for Physical-Chemical Processes, Italian National Research Council (IPCF-CNR), Messina, Italy. [4]Institute of Materials Science and Engineering (IMX), School of Engineering (STI), École Polytechnique Fédérale de Lausanne (EPFL), Lausanne, Switzerland. [5]Lausanne Centre for Ultrafast Science, École Polytechnique Fédérale de Lausanne (EPFL), Lausanne, Switzerland. [6]These authors contributed equally: S. Pullanchery, S. Kulik. ✉e-mail: ahassana@ictp.it; sylvie.roke@epfl.ch

ingredients of the classical continuum models by independent measurements.

As examples of this challenge, the following well-known pH-dependent phenomena can be considered: Pressure-driven electro-osmosis through hexagonal boron nitride nanocapillaries[15], carbon nanotubes[16], graphene oxide membrane conductance[17], and streaming currents through $MoS_2$ pores[18], all of which have vastly different surface chemistries in aqueous solutions. All these diverse experiments display a remarkable increase in current (x 2-3) when the pH of the surrounding bulk solution is increased from neutral to mildly basic (1 mM NaOH, pH 11). Hydrophobic nanoparticles, droplets, and gas bubbles of different materials all display the same pH-dependent change in electrophoretic mobility[19–25]. Increasing the pH of the surrounding aqueous bulk phase to the same mildly basic value results also in a x 2-3 increase in droplet mobility under the influence of an external electrostatic field. The involved poorly wetted/hydrophobic surfaces in the above examples are negatively charged. According to classical-continuum theory, this charge more than doubles under the influence of small changes in the bulk concentration of $OH^-$, increasing the current/mobility in an external electrostatic field[19,26] (due to the well-known electrostatic force of external field **E** on a charge q, **F** = q**E**). Curiously, the pH-induced increase in currents[15–18]/ mobility[19–25] is very similar in all of these cases, even though the substrates/systems are vastly different. It is well-known that ion-interface interactions are very sensitive to the ionic species and the chemical composition of the interface, but such effects manifest themselves at >0.1 M ionic strength[27], i.e. well above mildly basic conditions (<1 mM). Therefore, from a surface chemistry perspective, it is highly unlikely that dilute hydroxide bulk concentrations should lead to identical surface charging due to the surface activity of the hydroxide ions in this wide variety of systems.

Dispersion of oil nanodroplets in water is a well-studied model system for the hydrophobic/water interface where a pH-dependent increase in electrophoretic mobility has been observed. Using classical hydrodynamic models, numerous explanations for this pH-dependent electrophoretic mobility have been put forth: adsorbed $OH^-$, as it is naturally present in water and its concentration increases with pH[20], adsorbed $HCO_3^-$ from atmospheric $CO_2$[28] or surface-active impurities with titratable groups from either water or oil[29,30]. However, neither a spectral signature corresponding to hydroxide nor surface-active carboxylate impurities were detected on the surface in vibrational sum frequency scattering measurements[31–33], even though molecular groups with much lower cross-sections and in bulk concentrations as low as ~50 μM, were detected using this surface-sensitive method[34]. Recently, an explanation that does not involve ionic groups was proposed[32,33,35]: the local topological defects in the hydrogen (H-) bond network of interfacial water create charge density oscillations that transfer negative charge to the oil surface[35]. As a result of this interaction, the surface vibrational spectra of interfacial water exhibited a red shift in the O-D stretch modes and a blue shift in the interfacial oil C-H stretch modes. These frequency shifts revealed that charge transfer at the hydrophobic droplet surface involves the formation of improper H-bonds between water and oil[32]. Thus, water is the key ingredient that enables charge transfer and charge delocalization. To describe this type of charge, quantum-mechanical rather than classical model ingredients are required. It is, therefore, plausible that a classical-continuum description of charge is also insufficient to explain pH-induced increase in electrophoretic mobility. An increase in pH to mildly basic conditions is not expected to cause surface adsorption of $OH^-$ ions, but hydroxide ions can drastically modify the charge flow in the bulk. It is well known that charge flow of water's constituent ions, the proton and hydroxide, is enhanced by the Grotthuss mechanism[36]. Here, concerted H-bond network correlations involving the simultaneous breaking of hydrogen and covalent bonds contribute to charge displacement[36,37]. Correctly describing how such charge flows in the bulk modify the movement of negatively charged hydrophobic droplets also requires a quantum-mechanical description that explicitly includes electronic charge densities and their fluctuations. Moreover, the force exerted by an external electric field might further modify the electron density reorganizations. Therefore, determining the mechanism of pH-induced electrophoretic mobility requires several ingredients: (i) Independent measurements of surface potential and surface structure as a function of pH to determine if hydroxide ions modify the charge transfer at the droplet surface (ii) the influence of external electric field alone on the electron density reorganization around the hydrophobic droplet interface, and (iii) understand how the effects of electric field and the presence of hydroxide ions in the solution couple with each other to enhance the electrophoretic mobility of droplets.

Here, we embark on this route and show that a classical description of electrophoretic phenomena breaks down under the very general case in which the pH of the solution is varied and that, instead, a quantum-level description explicitly including partial charge transfer and electron density delocalization is needed to understand both bulk and surface behavior. We measured electrophoretic flow, electrostatic interface potential and molecular structure as a function of pH via three independent methods on the same hexadecane oil nanodroplet-water system. While exchanging the anion from $Cl^-$ to $OH^-$ in the aqueous bulk phase, a x2.2 increase in droplet mobility was recorded. However, the nanodroplets' electrostatic surface potential, the molecular surface structure, and the charge transfer of oil nanodrops and interfacial water were independent of the type of anion present in the solution. Simulating neopentane ($C_5H_{12}$) in water under an external electrostatic field, we find that the field induces a measurable electronic charge separation in the solute, and induces a net drift of neopentane. Investigating $Cl^-$ and $OH^-$ ions in neat water using quantum level ab-initio molecular dynamics (AIMD) simulations, we find that an external electrostatic field strongly modifies the charge distribution in bulk solution, leading to an enhanced charge density in basic solution, which is also distributed preferentially along the field lines. Droplet mobility, commonly seen as the mere result of the well-known force of an electrostatic field on ionic point charges, arises from an electrostatic field-induced gradient in the electrodynamic free energy that stems from a coupling between the electric field and the polarizability. Thus, quantum-mechanical charge fluctuations need to be explicitly included in the energetics/electrodynamics that underlies droplet mobility, and bulk (pH) responses.

## Results

### Electric field-induced droplet mobility

When oil nanodroplets or other small particles in water are subjected to an electrostatic field (**E**), the droplets move toward the positive electrode with a speed $v$, and a mobility $\mu = \frac{v}{E}$ (Fig. 1A). Classically, the mobility is converted to a ζ-potential, commonly interpreted as the electrostatic potential close to the surface due to the presence of ions/charged groups, via

$$\mu = \frac{\epsilon_0 \epsilon \zeta f(\kappa R)}{\eta} \qquad (1)$$

where $\epsilon_0$ is the vacuum permittivity, $\epsilon$ is the relative permittivity, $\eta$ is the viscosity of the medium and $f(\kappa R)$ is the Henry's function (Eq. 2, methods) that has a value between 2/3 and 1 depending on the radius $R$ of the droplet and the Debye screening parameter $\kappa$[1]. $1/\kappa$ represents the 1/e decay length of the electrostatic potential, which depends on the ionic strength of the solution.

We prepared hexadecane nanodroplets in water by ultrasonication (methods, Supplementary Fig. 1). Figure 1C shows the average ζ-potential values of 0.05 vol% hexadecane droplets in the presence of 1 mM NaCl (pH neutral, average radius 120 nm) and 1 mM NaOH (pH 11,

average radius 113 nm). Upon replacing $Cl^-$ with $OH^-$ ions, the mobility/ζ-potential magnitude increases by a factor of $2.2 \pm 0.2$. This dramatic pH-dependence is in excellent agreement with previous experiments (e.g., ref. [24]. from ~-51 mV at pH=6 to ~-100 mV at pH = 9, with ionic strength fixed at 1 mM).

## A pH-independent surface charge

Next, we determined the droplet surface potential ($\Phi_0$) using an all-optical method as a function of pH, and constant ionic strength. $\Phi_0$ is defined as the negative integral of the electrostatic field from the bulk aqueous solution up to the interfacial Gibbs dividing plane. $\Phi_0$ is retrieved from angle-resolved second harmonic scattering (SHS) measurements. In a non-resonant SHS experiment, a pulsed femtosecond (fs) near-infrared laser beam interacts with the droplets in solution (Fig. 1B). Coherent SH photons are emitted from all non-isotropic molecules that are anisotropically distributed. Since interfacial water is anisotropically distributed along the radial direction, while the bulk liquid is not, SHS has an exquisite interfacial sensitivity (Fig. 1B, right). Isotropically oriented molecules in bulk water display a weak but clearly visible SH emission that is not directional. We use these SH photons to calibrate the interfacial intensity, and measure the quantity $S(\theta)$. $S(\theta)$ represents the coherent interfacial SH intensity relative to the incoherent response of bulk water (Eq. 3, methods)[38]. $S(\theta)$ provides a calibrated intensity that is (1) comparable to any nanoparticle surface response, and (2) allows for calculating the surface potential ($\Phi_0$) and interfacial water ordering (quantified by the second-order susceptibility, $\chi^{(2)}_{s,2}$). Figure 1B (right) schematically illustrates the molecular origin of SHS response via $\chi^{(2)}_{s,2}$ and $\Phi_0$. By considering the combined interactions of the optical fields and the intrinsic interfacial electrostatic field with the optical properties of water, equations for $S(\theta)$ that connect $\Phi_0$ and $\chi^{(2)}_{s,2}$[38,39] can be derived. These equations are given in the SI (Supplementary Note 1, Eqs. S1-S2). Besides varying the scattering angle, it is also possible to change the polarization state of the optical beams. Of the four possible polarization combinations that

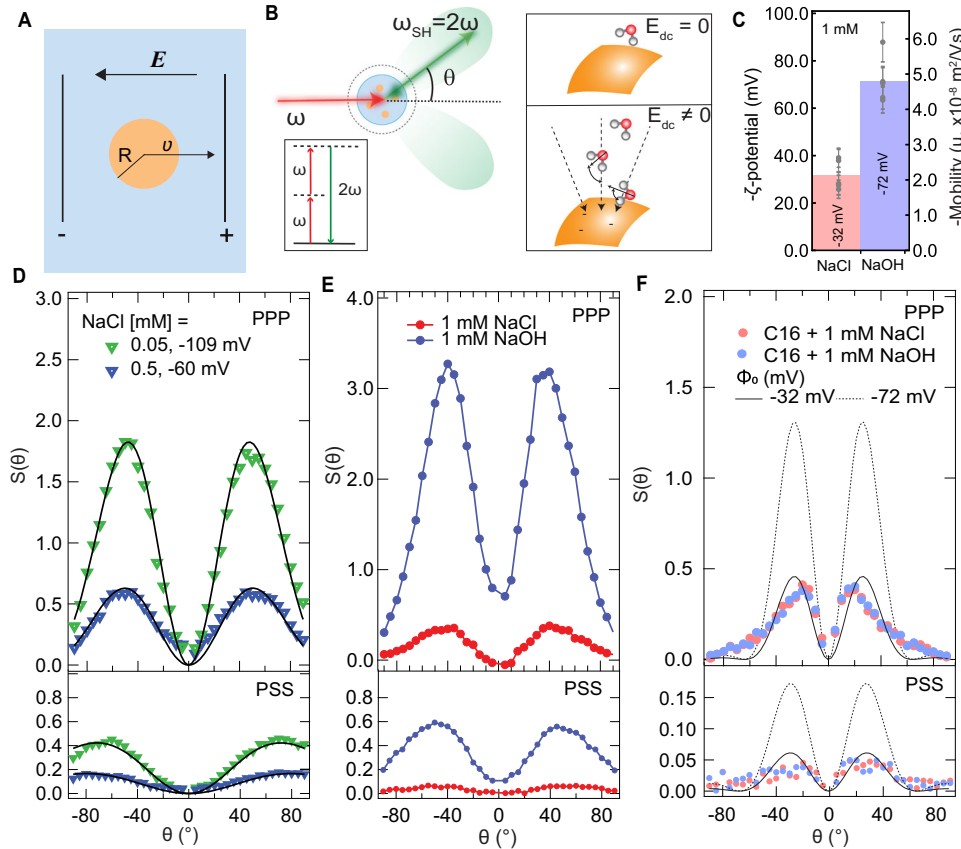

**Fig. 1 | Mobility is pH-dependent, but the surface charge is not. A** Illustration of an electrokinetic mobility ($\mu$) measurement. A droplet with radius R that has an interfacial charge is moved by an electrostatic field ($E$), resulting in a velocity $v = \mu E$. **B** Left: Angle-resolved second harmonic scattering. Near-IR fs laser pulses interact with droplet interfaces in a non-resonant SH scattering process. The SH photons are detected at different scattering angles, $\theta$. Inset: energy level scheme. Right: Schematic illustration of the origin of the SHS response. The SH response originates from water molecules anisotropically oriented at the interface. For a neutral interface with the electrostatic field, $E_{dc} = 0$, only the water molecules immediately near the surface are anisotropically oriented and contribute to the SH intensity. For a charge interface, the non-zero electrostatic field penetrates into subsequent layers of water and reorients the water dipoles along the field lines. **C** The average ζ-potential values of 0.05 vol%-120 nm radius hexadecane droplets in 1 mM NaCl (red bar) and 1 mM NaOH (blue bar) solutions, computed using Eqs. 1 and 2. The gray dots are data points measured from 5 different samples, and the error bars denote the standard deviation of 3 measurements performed on each sample. The average mobility values are $\mu = -2.2 \cdot 10^{-8}$ m²/Vs (NaCl) and $\mu = -4.8 \cdot 10^{-8}$ m²/Vs

(NaOH), respectively. **D** Angle-resolved SHS patterns, measured in both PPP (top) and PSS (bottom) polarization combinations, obtained from 0.2 vol%, 120 nm diameter 1,2-dioleoyl-sn-glycero-3-phospho-L-serine (DOPS) liposomes in aqueous solution, containing 0.05 mM (green data) and 0.5 mM (blue data) NaCl. The surface potentials of the liposomes are −109 mV (green data) and −60 mV (blue data) respectively. PPP refers to all beams polarized in the horizontal scattering plane, and PSS refers to the SH beam polarized in the scattering plane, and the fundamental beam polarized perpendicular to it. **E** Angle-resolved SHS patterns, measured in both PPP (top) and PSS (bottom) polarization combinations, obtained from 0.1 w.w.% 300 nm diameter silica nanoparticles in aqueous solution with 1 mM NaCl (red) and 1 mM NaOH (blue). **F** Measured SH $S(\theta)$ scattering patterns of 0.05 vol% hexadecane droplets in 1 mM NaCl (red data) and 1 mM NaOH (blue data) using PPP and PSS polarization combinations. The black lines represent calculated SH $S(\theta)$ patterns in both polarization combinations, having $\Phi_0 = −32$ mV (solid line) and $\Phi_0 = −72$ mV (dashed line), see supplementary Note 1, Eqs. S1 and S2, and Supplementary Tables 1–4 for method, equations, and parameters.

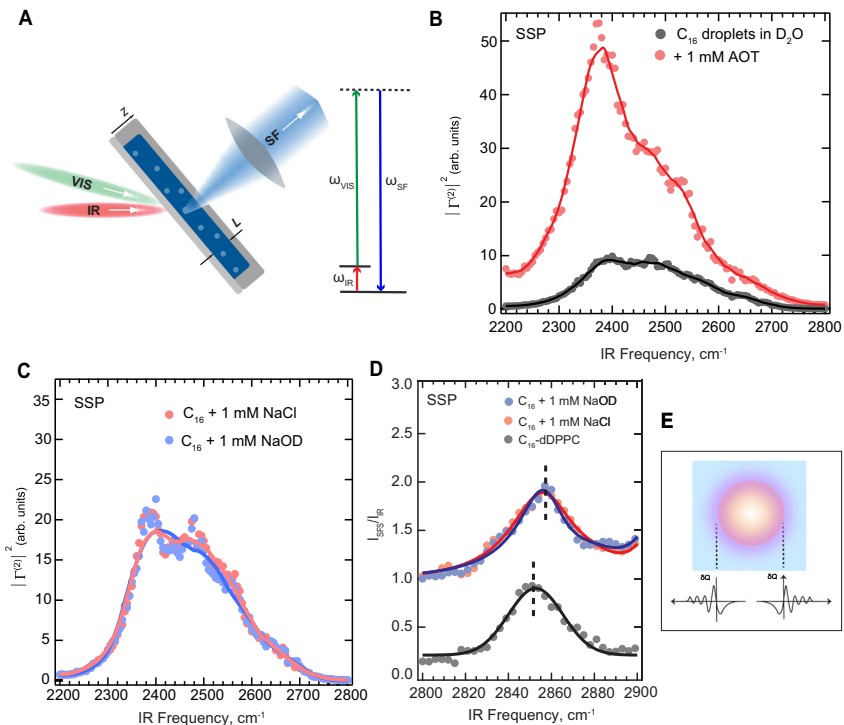

**Fig. 2 | H-bonding network and charge transfer mechanism are not pH-dependent. A** Vibrational sum frequency scattering: Simultaneous excitation using pulsed fs IR and VIS laser beams results in a simultaneous IR and Raman excitation that gives rise to scattered sum frequency (SF) photons that report on the interfacial molecular vibrational spectrum. **B** The O-D stretch spectrum of 2 vol%100 nm radius hexadecane droplets in pure $D_2O$ (black) and in the presence of 1 mM sodium bis(2-ethylhexyl) sulfosuccinate (AOT) surfactant, measured with the SSP polarization combination (SF and VIS beams polarized perpendicular to the scattering plane, IR beam polarized parallel to the scattering plane). See "Methods" for experimental details, supplementary Note 2 and Supplementary Figs. 2 and 3 for details on retrieving the O-D stretch spectra. **C, D** Vibrational SFS spectra of O-D stretch modes (**C**) and C-H stretch modes (**D**) of 2 vol% hexadecane droplets in $D_2O$ at pD 7 (1 mM NaCl, red) and pD 11 (1 mM NaOD, blue) recorded using the SSP polarization combination). The gray curve shows C-H stretch modes of oil droplets that are not interacting with water, achieved by covering them with a dense monolayer of deuterated DPPC lipids. Solid lines in **B**–**D** represent the moving average as a guide to the eye, as described in methods. Note that the data in **C, D** are not normalized. **E** Schematic illustration of charge distribution at the interface of an oil nanodroplet as a consequence of improper H-bonding/charge transfer effects. The surface charge density in the oil phase in contact with pH neutral water as arising from charge-transfer effects was computed to be $\sim-0.015$ e/nm², ref. 35.

lead to SH emission from non-chiral interfaces, two independent polarization combinations (PPP and PSS) remain. Having 2 sets of scattering patterns with 2 unknowns, $\chi^{(2)}_{s,2}$ and $\Phi_0$, can be determined unambiguously for a given droplet size distribution, without making any modeling assumptions on the surface structure/surface electrostatics. For particles <300 nm in radius, this method is exceptionally sensitive to changes in either the surface chemical structure (via $\chi^{(2)}_{s,2}$) or the electric double layer (via $\Phi_0$), especially for electrolyte concentrations <10 mM[38,40,41]. To illustrate this exquisite sensitivity, Fig. 1D shows SHS $S(\theta)$ patterns recorded from aqueous solutions of charged liposomes (obtained from the same stock solution) having $\Phi_0$ values that differ by a factor of ~2 (changing between −60 mV and −109 mV). The SHS patterns vary drastically in shape and intensity (x 3). Figure 1E shows SHS patterns from silica nanoparticle dispersions containing 1 mM NaCl (red data) or 1 mM NaOH (blue data). The $S(\theta)$ patterns, recorded from dispersions having the same number of particles with the same size distribution, are ~10 x different in intensity. Both these examples demonstrate the unique sensitivity of the SHS experiment to surface potential (Fig. 1D) and surface structure and surface potential (Fig. 1E). Therefore, if replacing 1 mM NaCl by 1 mM NaOH in a solution of oil droplets in water, would result in a change in surface potential by a factor of 2.2 ± 0.2, this should be reflected in a drastic change of the SHS pattern.

Figure 1F shows the measured SH $S(\theta)$ patterns of 100 nm radius oil droplets in 1 mM NaCl (red) and 1 mM NaOH (blue), having ζ-potential values of −28 ± 4 mV and −88 ± 11 mV, respectively. The two scattering patterns obtained in NaCl and NaOH solutions are indistinguishable. The black solid line is a computation of the SHS scattering response $S(\theta)$ with $\Phi_0 = -32$ mV, which describes the data reasonably well. Increasing the surface potential, while keeping all other parameters the same, results in the black dashed line, which is ~2.8x higher in intensity. Clearly, the measured data does not show a difference between both samples. Thus, for oil droplets in water, the interfacial properties $\Phi_0$ and $\chi^{(2)}_{s,2}$ are not changing when $Cl^-$ is replaced by $OH^-$ in the bulk aqueous phase. Increasing the pH from neutral to pH = 11 at identical ionic strength does not change the surface charge density. Thus, the traditional explanation that a drastic change in mobility/ζ-potential data derives from changes in the ionic surface charge density or ion distribution in the electric double layer needs to be reconsidered.

## pH-dependence of the interfacial H-bond network and charge transfer

In order to determine the interfacial H-bond and oil C-H structure as a function of pH, we used vibrational sum frequency scattering (SFS) spectroscopy. Vibrational SFS relies on the interaction of an infrared (IR) and visible (VIS) beam with the droplet dispersion (Fig. 2A). SF photons are scattered from regions where the molecules in the otherwise isotropic solutions are anisotropically arranged, i.e., they are scattered from the interface of the droplets reporting on the interfacial

structure of the hexadecane and water molecules. Like SHS, SFS is exceptionally sensitive to surface chemistry. The interfacial water response, as in SHS, reports on both the surface structure (via $\chi_{s,2}^{(2)}$) and the surface potential ($\Phi_0$) according to similar expressions that are now frequency-dependent and include a vibrational resonance.

The C-H stretch region reports on the chain conformation of alkyl chains, while the O-D stretch region reports on both the surface potential (via the intensity) and the surface H-bond network structure (via the spectrum). Figure 2B shows the effect of adding 1 mM of surface-active compound (sodium bis(2-ethylhexyl) sulfosuccinate (AOT)) to the aqueous solutions of oil nanodroplets in water for the O-D stretch region. The effect this has on the C-H stretch region is shown in Supplementary Note 3 and Supplementary Fig. 4. As with SHS, adding 1 mM leads to big changes (x 5) in SFS intensity, supplemented by changes in spectral shape. Thus, nanodroplet surface adsorption, even at bulk millimolar concentrations, results in drastic changes in vibrational SFS spectra.

Figure 2C shows SFS $|\Gamma^{(2)}|^2$ spectra in the O-D stretch region obtained from 100 nm radius hexadecane droplets dispersed in $D_2O$ solutions with 1 mM NaCl (pH neutral, red) and 1 mM NaOD (pH = 11, blue). The methods section contains information about the sample preparation[42] and the SFS method[32,43]. Figure 2D shows spectra recorded in the C-H stretch region, which reports on the interfacial oil. For the interfacial water, the SF spectra obtained at pD 7 and pD 11 consist of previously identified features[32,44]: Two broad features around 2395 cm$^{-1}$ and 2500 cm$^{-1}$ that correspond to H-bonded water molecules at the interface, with the latter being more weakly H-bonded compared to the former. Approximately half of the spectral broadening arises from vibrational coupling[44]. The shoulder above ~2600 cm$^{-1}$ was attributed to O-D bonds that do not form H-bonds with other water molecules, but rather participate in weak improper H-bonds with the C-H groups of the oil. These improper H-bonds are responsible for the transfer of a tiny amount of charge from water to oil (computed to be ~0.015 e-/nm$^2$)[35], which generates the negative charge on the oil droplets[32], responsible for their stability (Fig. 2E). Interestingly, the spectrum recorded in the presence of 1 mM NaOD (pD 11, Fig. 2C, blue) is indistinguishable from the one recorded in the presence of the 1 mM NaCl (pD 7, Fig. 2C, red). This implies that OD$^-$ and Cl$^-$ ions have identical impacts on the interfacial H-bond network. Thus, there are no pH-induced differences in the H-bond network at the interface between oil droplets and water.

The SFS spectra of the interfacial C-H modes (Fig. 2D) show both the symmetric CH$_2$ (CH$_2$-ss) and CH$_3$ (CH$_3$-ss) stretch modes around ~2850 cm$^{-1}$ and ~2878 cm$^{-1}$, respectively. Spectra are recorded for bare oil droplets in pD 7 (red data) and pD 11 (blue data) water, and compared to oil droplets that were covered with a monolayer of deuterated 1,2-dipalmytoil-sn-glycero-3-phosphocholine (d-DPPC, 0.48 nm$^2$/lipid[45], gray data), a zwitterionic lipid. The purpose of the lipid monolayer is to shield the oil from interacting with the water, and to simultaneously remove the interfacial charge[32]. The oil CH$_2$-ss modes adjacent to the deuterated lipid alkyl tails of the charge neutral droplets vibrate at lower frequencies (2851 cm$^{-1}$) than the CH$_2$-ss modes of the charged bare oil droplets that are adjacent to pH neutral water (2856 cm$^{-1}$). This frequency shift correlates with the charge on the droplets and is a signature of charge transfer between water and oil (Fig. 2E)[32]. We use the gray spectrum here as a reference, to identify whether increasing the pH results in a higher frequency shift, and thus an increase in charge transfer. Comparing the C-H mode spectra recorded at pD 7 and pD 11 in Fig. 2C, no additional frequency shift is observed at higher pH. Therefore, Figs. 2C and 2D show that neither the interfacial structure of the H-bond network of water, nor the amount of charge transfer is pH-dependent. We also note that, had there been adsorption of OH$^-$ or other ions, this would have led to significant changes in the SHS scattering patterns and the SFS spectra[31] (See Supplementary Note 4 for more details on ions/impurities and

how their role is eliminated in our findings). Yang et al.[46] observed that the sum frequency O-H stretch spectra at a planar pristine hexane/water interface increased in overall intensity with increased pH. The differences between this study and our present results might originate from either the difference in the model system (nanospheres vs extended planar interface) or the response of the hexane, as detailed in the Supplementary Note 4. Further studies might be required to reconcile the differences. Nevertheless, the absence of interfacial OH$^-$ that we observe agrees with a recent Raman MCR study that showed OH$^-$ being excluded from the hydration shells of small hydrophobic molecules (with pH > 12)[47]. Thus, we observe that interchanging Cl$^-$ and OH$^-$ does not result in additional surface charge (Fig. 1) or in a change of interfacial interactions (Fig. 2). Furthermore, the oil and water surface structure are identical for both cases. Therefore, an increase in droplet mobility (Fig. 1C) cannot originate from an increase in surface charge density, as suggested by considering only ionic species as charge carriers within the classical theory. To explain the experimental observations (pH-dependent mobility without an increase in surface charge), we turn to first principles simulations in order to provide some microscopic signatures of the phenomena observed.

## The effect of an external electric field on hydrophobic objects in water

To rationalize the mechanism by which a neutral hydrophobic object moves in the presence of an external electrostatic (i.e., static and homogeneous) electric field, we turn to examining the electronic properties of a single neopentane molecule in a box of water molecules using Density Functional Theory (DFT) based ab initio molecular dynamics (AIMD). We have shown in previous work[32,35] that improper H-bonds forming between C-H groups of dodecane and water lead to a transfer of charge between water and oil, leaving the oil phase in contact with water negatively charged. We expect that such electron density fluctuations play a key role in explaining the mobility. Studying the coupling between hydrodynamics and electronic fluctuations is currently limited by challenges such as finite size and simulation timescales affordable, as well as the quality of the underlying electronic structure description. Our efforts in this regard are aimed at providing some initial features that we believe warrants further theoretical investigation.

We begin by showing in Fig. 3A the change of the Hartree potential as obtained from the simulations in the presence of an applied oriented external field. The electric field induces a sizable asymmetry in the potential across the neopentane molecule along the field direction, as indicated by the purple isolines in Fig. 3A. This type of electronic polarization is, by construction, not included in any classical description used to rationalize hydrodynamic flow. To illustrate this point further, we compared the evolution of the dipole moment of hydrated neopentane obtained from the AIMD simulations in the presence of different field strengths. As displayed in Fig. 3B, the electric field significantly polarizes the neopentane molecule in water as witnessed by the progressive growth of the dipole moment from 0.5 to 1.5 D. The dipole moment of the water molecules also exhibits subtle changes albeit much smaller compared to that of neopentane. These results point to the important role of electronic polarization enhanced by the external field on simple hydrocarbons in water.

The preceding analysis is performed with standard DFT functionals which are known to show some issues of electronic over-delocalization due to the well-known self-interaction error and the related delocalization error[48–50]. In order to validate the features we observe, namely the importance of electronic polarization and charge transfer in the interactions between neopentane and water, we performed many-body absolutely localized molecular orbital energy decomposition analysis[51–54] calculations using the $\omega$B97M-V[55,56] density functional with the aug-cc-pVTZ[57] basis set. Figure 3C shows the average 2-body (2B) polarization (reorganization of electron density

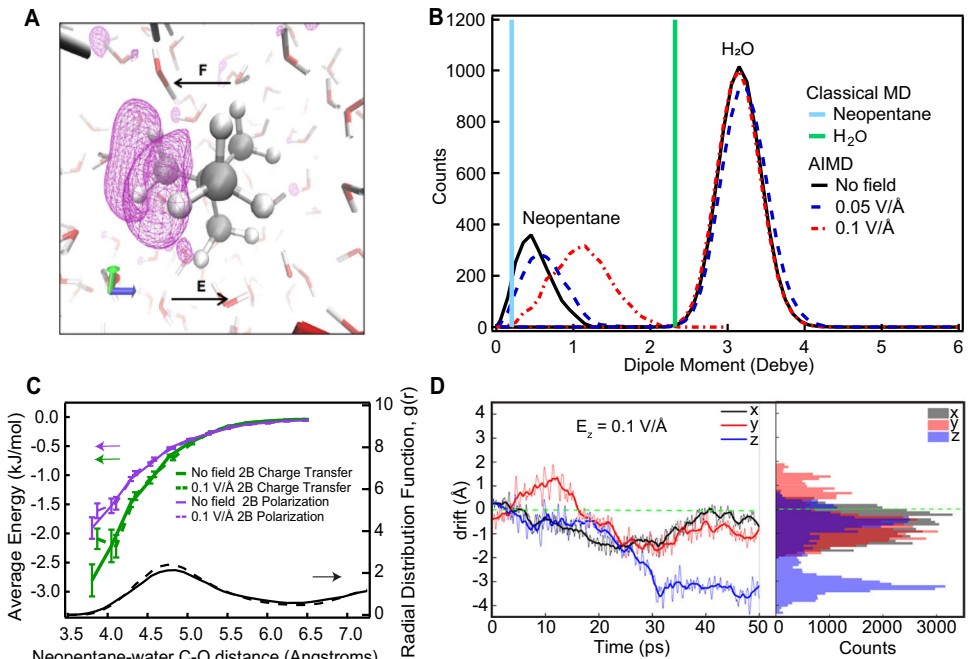

**Fig. 3 | Charge transfer effects contribute to mobility of a hydrophobic molecule in water. A** Change of the Hartree potential (iso-value equal to 0.002 e·*Ha/Å) of neopentane when an electrostatic field of magnitude 0.1 V/Å is turned on in the z-direction. The resulting force is indicated with an arrow. **B** Distribution of dipole moments in the solution for both the classical and quantum-mechanical representations. Note that due to the high-level requirements on the computations in panels **B**, **C**, we employ a field strength that is significantly higher than in the experiment, but well below any threshold for chemical modification[72], as described in the methods section: Ab initio molecular dynamics simulations. **C** 2-body (2B) interaction energy per water molecule between the water and neopentane separated into the contributions coming from the polarization and charge-transfer. The solid and dashed lines correspond to the average energy contributions of molecular geometries taken from simulations in the absence of an external electric field and in the presence of a 0.1 V/Å field, respectively (note that all polarization and charge transfer energies were calculated without any external field). The error bars correspond to the standard error. **D** Cartesian components of the drift (dashed lines along with their running averages: solid thicker lines) of the C atoms of the neopentane molecule under the action of a 0.1 V/Å field oriented along the z-axis. Negative values of the z-component of the drift (magnified in the blue histogram on the right) give further evidence of the importance of mobile electron density fluctuations on neopentane and in water in the presence of an external field and are the consequence of the force shown in panel **A**.

inside the neopentane) and charge transfer (transfer of electron density from water to neopentane via improper H-bonds) contributions to the interaction energy between water and neopentane as a function of the neopentane central-carbon and the water oxygen distance. The solid and dashed curves show decompositions for geometries sampled from the no-field simulation, and the 0.1 V/Å simulation, respectively. All ALMO EDA calculations were performed without an external field. We observe that the interaction energy for both of these effects is indeed significant with the charge transfer being slightly larger than the polarization. The 2-body decomposition shown here in Fig. 3C strongly suggests that the dipole moment picked up by the neopentane involves a combination of both charge transfer from the water to the molecule as well as internal reorganization of the electron density associated with the intrinsic polarization tensor of neopentane. The remaining terms in the decompositions as well as estimates of the 3-body polarization and charge transfer energies are shown in the Supplementary Note 5 and Supplementary Fig. 5.

### Electric field-induced flow of neopentane

An external electric field induces anisotropic electronic charge separations in hydrophobic molecules (Fig. 3A–C) when they are considered on the quantum-mechanical level. Such a field-induced anisotropic depletion/accumulation of the electron density in favor/against the field axis has to be considered in conjunction with the spontaneous (i.e., in the absence of the field) near-isotropic charge transfer from the solvation water molecules to the solute[32,35], also mentioned in the previous sections. In this context, the former effect may be thought to enhance the asymmetry in charge transfer on either side of the oil-droplet or equivalently create an asymmetric distribution of local dipoles created

at the oil-water interface. Having induced a charge separation upon applying the external field, we note that this constitutes a polarization, **P**. This means there will be an electric-field-induced energy ($U$) gradient as well and thus a force in the direction of the external field ($\mathbf{F}_z = -\frac{dU}{dz}$) emerges naturally. Namely, when an external electrostatic field ($\mathbf{E} = \mathbf{E}_z$) interacts with polarization fluctuations, **P**, there will be local currents ($\mathbf{J} = \frac{d\mathbf{P}}{dt}$), and thus electrodynamic work ($\int (\mathbf{J}.\mathbf{E})d\mathbf{r}^3$) is done[58]. This force can be verified in our simulations by comparing the speed of the neopentane with the external field on (Fig. 3D) and field off (Supplementary Fig. 6). Indeed, the neopentane drifts in the opposite direction of the field (while it does not do so when the field is off). This is consistent with the neopentane acquiring an effective net negative charge of −0.05e⁻ (using a Mulliken charge scheme) from the interaction with water. We note that the exact magnitude of the charge on the neopentane will obviously be sensitive to the charge partitioning scheme used. However, it is clear that the drift of the neopentane molecule in the presence of the field resembles that of an effectively negatively charged object.

As alluded to earlier, the field strengths used in the simulations are rather large in magnitude (0.1 V/Å) in order to capture the observed flow on realistic timescales. Nonetheless, our simulations yield a mobility $\mu \sim -1 \cdot 10^{-8}$ m²/Vs and a corresponding ζ-potential of ∼−30 mV of the neopentane molecule. Both these numbers are in fairly good agreement with the experiment (Fig. 1C) despite the theoretical and computational limitations.

### Polarization and the Grotthuss mechanism

Our experiments show that the mobility/ζ-potential of the oil droplets are enhanced in basic compared to saline solution. The surface sensitive measurements demonstrate that this enhanced mobility cannot be

enhanced due to specific binding of the ions to the surface of the droplet. We propose that the effect observed is rooted in a fundamental difference between the mechanisms of charge transport of $OH^-$ vs $Cl^-$ ions in the bulk. $Cl^-$ ions are symmetrically hydrated, while

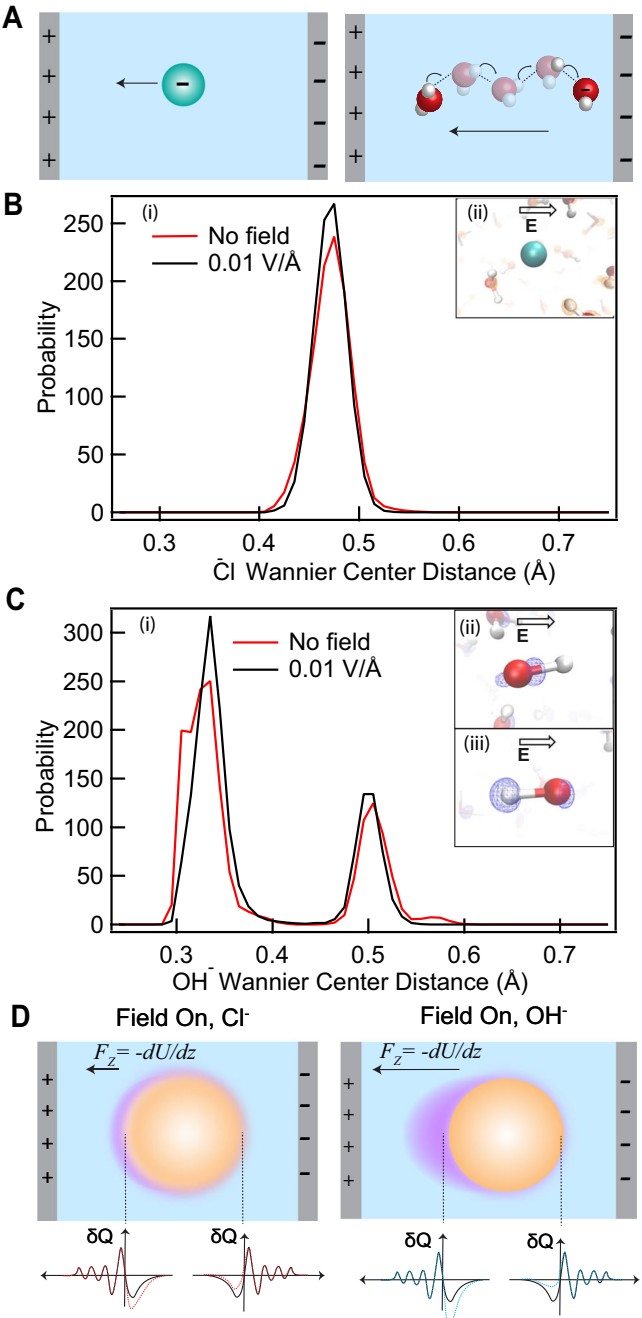

**Fig. 4 | pH modifies charge distribution in bulk aqueous solutions. A** Illustration of the difference between charge transport in a bulk solution containing chloride (left, hydrodynamic flow), or hydroxide ions (right, Grotthuss mechanism, correlated H-bonds). **B** Wannier Center-$Cl^-$ distance distribution shown in black and red with and without field, respectively (i), and snapshot of the response of the electron density around $Cl^-$ upon application of an external electric field (ii). **C** Wannier Center-$OH^-$ distance distribution shown in black and red with and without electric field, respectively (i), and snapshot of the response of the electron density around $OH^-$ upon application of an external electric field along (ii) and against (iii) the OH covalent bond. The arrows represent the direction of the electrostatic field. **D** Schematic illustration of change in charge distribution on an oil nanodroplet in water when an external electrostatic field is applied (dashed lines, left). (right): Additional asymmetry is induced due to the difference in polarization of the bulk basic solution in the presence of an external electrostatic field.

$OH^-$ ions are solvated asymmetrically. The $OH^-$ ion is a topological H-bond network defect that has a hydrophilic side that accepts up to 4 H-bonds and a more hydrophobic side that donates a weak H-bond[59]. Thanks to these defect structures, the H-bond network itself becomes a more efficient conduit for charge transport, with charge transferring via the Grotthuss mechanism. This involves protons moving through the H-bond network by means of the interconversion of covalent and H-bonds[37]. This mechanistic difference in charge transport is schematically depicted in Fig. 4A. As a consequence of this difference, dilute $OH^-$ solutions conduct charge x ∼ 2 faster than dilute $Cl^-$ solutions: The molar conductivity, $\Lambda_m$, is 244.60 Scm²/mol for 1 mM NaOH in water and 123.68 Scm²/mol for 1 mM NaCl, even though the dielectric constant and viscosity of both solutions are identical[60]. Given a certain applied electrostatic field (**E**) to a solution with a charge conductivity $\sigma$, the current density $\mathbf{J} = \sigma \cdot \mathbf{E}$ is therefore x ∼ 2 bigger, which means that the solutions' polarization **P** must also be bigger (since they are related via $\mathbf{J} = \frac{d\mathbf{P}}{dt}$). To understand the effect of an electrostatic field on bulk aqueous solutions of $OH^-$ and $Cl^-$, we next investigate the charge distribution in hydroxide and chloride bulk aqueous solutions under the influence of an electrostatic field using quantum level AIMD simulations.

### Electronic structure of $OH^-$ vs $Cl^-$ in aqueous solution

To visualize the electron density around hydroxide and chloride ions in water, we computed the Wannier centers of the $Cl^-$ and $OH^-$ ions in liquid water, with and without the application of an external field of $10^8$ V/m (i.e., 0.01 V/Å) (more details in methods).

Figure 4B, C compare the distributions of the Wannier centers with respect to the Cl nucleus in the case of the $Cl^-$ and with respect to the O nucleus of the $OH^-$ ion, shown with and without the field in black and red, respectively. In the case of the $Cl^-$ ion, the presence of a single peak illustrates the spherical symmetry of the local electron density, while the $OH^-$ ion is characterized by two different sets of Wannier orbitals. For the $OH^-$ ion, these two Wannier orbitals correspond to the lone pair electrons and those residing along the covalent bond shared between the oxygen and the proton. Comparing hydrated $Cl^-$ and $OH^-$, we find that an external electrostatic field increases the amount of locally displaced charge density in $OH^-$ solutions more than in $Cl^-$ solutions. Furthermore, $OH^-$ solutions display a large amount of charge density asymmetry in the direction of the field, while for $Cl^-$ solutions there is no field-induced directionality, consistent with the asymmetric response of the local H-bonds observed for $OH^-$ solutions. Thus, at identical ionic strengths, external fields induce a different response in the electronic polarization for pH neutral and basic solutions.

### pH modifies electronic polarization of the bulk solution which impacts droplet mobility

Turning now to the oil droplet motion, we can explain both its stability, its mobility and why the pH change leads to more than a duplication of the mobility (Fig. 1C), while at the same time surface charge and charge transfer of a whole droplet are not impacted (Figs. 1F and 2C, D). In the absence of an electrostatic field, the droplet interface experiences electron charge density fluctuations that originate from H-bond dynamics at the interface (both from improper C-H...O bonds, and O-H...O bonds, Fig. 2E). This results in a net negative charge, which ensures that the droplets are stable against coalescence. In addition, based on the results of Figs. 3 and 4, we propose that an external electrostatic field changes the charge distribution on either side of the droplet interface anisotropically, somewhat depleting the negative charge on the side of the negative electrode, and enhancing it on the side of the positive electrode (Fig. 4D). Thus, the nanodroplet has a negative charge-transfer-induced net charge and a field-induced anisotropic charge density displacement. The external field acts on both, resulting in a force in the opposite direction of the field axis. In analogy

to the neopentane case, there is an electrodynamic polarization and an energy gradient, which constitutes such a net force. In basic solutions, the charge conductivity is larger (x 2). This increase in conductivity is a consequence of the Grotthuss mechanism and the charge anisotropy it enables. Being a more conductive medium to the flow of negative charge, we suggest that there will be an equally larger droplet mobility, as the external electric field exerts a force on the negative charges on the droplet. Because the charges move 2 times faster, so does the droplet. Unraveling the molecular mechanism by which the Grotthuss mechanism facilitates faster motion of the oil-droplet warrants further study.

## Discussion

Numerous different experiments involving liquid flow along charged surfaces display very similar pH-dependent effects, despite their surface chemistries being wildly different. Here, we provided a general framework for rationalizing the origins of these phenomena. A classical continuum model description of electrophoretic phenomena breaks down under the very general case when: (i) Interfacial charge transfer effects become important, i.e. on poorly wetted surfaces in water, and (ii) the pH of the solution is varied, involving the Grotthuss mechanism. This warrants a deeper investigation into the electro-hydrodynamics of hydrophobic objects in water as a function of pH. We, therefore, measured the electrophoretic mobility of hydrophobic nanoscale oil droplets in water and combined it with nonlinear optical spectroscopic measurements as a function of pH. Surface potential and charge were measured as a function of pH under conditions of constant ionic strength whereby $Cl^-$ and $OH^-$ were interchanged. We also used vibrational sum frequency scattering to determine the pH-dependent molecular interfacial structure and interactions. While the mobility increased drastically upon increasing the pH, neither did the surface potential, nor the interfacial interactions. In other words, the surface charge turned out to be pH-independent.

Ab initio molecular dynamics (AIMD) simulations of neopentane in water ('the smallest oil droplet") with an external electric field show an electronic charge polarization within the molecule, which is proportional to the external field strength. Applying an electrostatic field across solutions of $OH^-$ and $Cl^-$ ions using AIMD leads to a different polarization, with an electron density distribution that is both enhanced and anisotropic for $OH^-$, but not for $Cl^-$. Besides the translational drift of the neopentane in the presence of field, visual inspection of our simulations also indicates that the molecule also undergoes rotational motion. This effect is likely due to the anisotropic response of charge in the presence of the field. The coupling of these translational and rotational motion is another aspect that warrants further study.

Combining experimental findings and quantum-level considerations, we arrive at a description of an electric-field-induced net force that originates from gradients in the fluctuating electric polarizability of the droplet surface, which ultimately arises from electron density fluctuations leading to charge transfer. Although it is well known that charge gradients in external fields – or field gradients with fixed charge distributions – give rise to forces, as e.g. manifested in dielectrophoresis[61] or electrorotation[62], the force discussed here cannot result in any way from a purely classical continuum treatment, as the charges in such a description are not permitted to be delocalized across different molecules, and would therefore not display the type of behavior that one finds using a quantum-mechanical approach. At the same time, our experiments clearly indicate that the surface charge and ion distribution in the electric double layer is not changing as a function of pH. The important role of electronic polarization as well as that of the Grotthuss mechanism, indicates that one must go beyond classical continuum models.

The induced charge density gradient interacts with the external electric field itself and results in a force, as it represents a gradient in the electrodynamic energy distribution. This force is a more complete rendering of F=qE, considering not only classical charges but also dynamic fluctuations of charge density displacements. As this force acts on the charges of the droplet, the droplet moves. On neopentane, the field that acts on the induced dynamic polarization induces a mobility of the same order of magnitude as the experimentally measured mobility of the nanodroplets. In basic solutions, the global charge asymmetry/polarization gradient is enhanced by ~x 2 because the charge conductivity (and thus current density/change in polarizability) is larger by exactly the same amount, and it couples to the electrodynamic energy. In the case of gas-bubbles, the charge transfer mechanism involves only the water molecules close to the surface which leads to an effective negative charge. This mechanism is very general and impacts many processes in biology, chemistry, and nanotechnology and provides an explanation of why bulk acidity or basicity can have important implications in transport phenomena.

## Methods

### Chemicals

Hexadecane (highest available analytical standard, Sigma Aldrich, <5 mL vials), $D_2O$ (99.8% atom % D, Acros Organics), sulfuric acid (95–97%, Merck), $H_2O_2$ (30%, Reactolab SA), $d_{75}$-DPPC (Avanti Polar Lipids), NaCl (99.999%, Acros), NaOH (99.99%, Sigma-Aldrich), and NaOD solution (40 wt.% in $D_2O$, 99.5 at.% D, Sigma-Aldrich) were used as received. The purity of hexadecane was verified with a Zisman test[63]. The glassware used to prepare and store the nanodroplets was freshly taken out of the manufacturer's packaging and never reused after the preparation. As a first step, the glassware was cleaned with a freshly prepared piranha (3:1 $H_2SO_4$:$H_2O_2$) solution. After being immersed in the piranha solution for ~45 minutes, the glassware was rinsed copiously with ultrapure water (18.2 M$\Omega$·cm), obtained from a Milli-Q UF-Plus instrument (Millipore Inc.)

### Oil nanodroplets in water

As described previously[42], mixtures of 2 vol% hexadecane in pure $H_2O$ or $D_2O$ were first mixed for ~3 minutes using a vortexer (IKA® Vortex 2). These mixtures were then ultrasonicated (35 kHz, 400 W, Bandelin) until monodisperse droplets with diameters in the appropriate size range were formed, and diluted as required by the different experiments. The size distributions of the nanodroplets were characterized by dynamic light scattering (DLS) using a Malvern Zetasizer Ultra instrument. The size distribution of hexadecane droplets in water with 1 mM NaCl and 1 mM NaOH is shown in Supplementary Fig. 1. Oil droplets covered with d-DPPC (Fig. 2D) were prepared by mixing 2 vol % hexadecane with $D_2O$ containing 1 mM lipid. The mixture was then homogenized for 2 min with a hand-held homogenizer (TH, OMNI International) using an angular velocity of 15 rpm and subsequently placed in an ultrasonic bath (35 kHz, 400 W, Bandelin) until monodisperse droplets were formed.

### Electrokinetic mobility measurements

The electrophoretic mobility measurements were performed using laser Doppler velocimetry and phase analysis light scattering, employing a dynamic light scattering instrument (Malvern Zetasizer Ultra). To perform the electrophoretic mobility the nanoemulsions were diluted to 0.05 vol% by adding ultrapure water. The electrophoretic mobility ($\mu$) values were converted into $\zeta$-potential ($\zeta$) values using the following expression: $\mu = \frac{\epsilon_0 \epsilon \zeta f(\kappa R)}{\eta}$ where $\epsilon_0$ is the vacuum permittivity, $\epsilon$ is the relative permittivity of water (78.17), $\eta$ the viscosity of water (0.89 cP), $f(\kappa R)$ is Henry's function, $\kappa$ is the inverse Debye length ($\sqrt{\epsilon_0 \epsilon k_B T / 2.10^3 N_A e^2 I}$, where $k_B$ is the Boltzmann constant, $T$ is the temperature, $N_A$ is the Avogadro's number, $e$ is the elementary charge and $I$ is ionic strength), and $R$ is the radius of the droplet. The typical approximations used for Henry's function are the Smoluchowski formula in which $f(\kappa R) \to 1$, and the Hückel formula in

which $f(\kappa R) \to 2/3$. In the case of oil droplets in water, we use a more generalized form proposed by Oshima[64] for Henry's function:

$$f(\kappa R) = \frac{2}{3}\left[1 + \frac{1}{2\left\{1 + \frac{2.5}{\kappa R(1 + 2e^{-\kappa R})}\right\}^3}\right] \tag{2}$$

Note that this assumes no slip as a boundary condition.

## Angle-resolved second harmonic scattering

SHS measurements were performed using 190-fs laser pulses centered at 1028 nm with a 200 kHz repetition rate. The polarization of input pulses was controlled by a Glan-Taylor polarizer (GT10-B, Thorlabs) in combination with a zero-order half-wave plate (WPH05M-1030). The filtered (FEL0750, Thorlabs) input pulses with a pulse energy of 0.3 μJ (incident laser power $P = 60$ mW) were focused into a cylindrical glass sample cell (inner diameter 4.2 mm) with a waist diameter of ~35 μm and a Rayleigh length of 0.94 mm. The scattered second harmonic light was collected with a plano-convex lens ($f = 5$ cm), and then filtered (ZET514/10x, Chroma), polarized (GT10-A, Thorlabs), and finally focused into a gated photomultiplier tube (H7421-40, Hamamatsu). The angle of acceptance for the signal collection was 3.4°. The scattering pattern was measured at a scanning step of 5°, between -90° $< \theta <$ 90°. Each data point was acquired with an acquisition time of $20 \times 1$ s and a gate width of 10 ns. All measurements were performed in a temperature- and humidity-controlled room ($T = 297$ K; relative humidity, 26.0 %). The normalized SHS intensity $S(\theta)$ at the angle θ was calculated as:

$$S(\theta) = \frac{I(\theta)^{PXX}_{sample} - I(\theta)^{PXX}_{solvent}}{I(\theta)^{SSS}_{H2O}} \tag{3}$$

where $I(\theta)^{PXX}_{sample}$ and $I(\theta)^{PXX}_{solvent}$ are the average SHS intensities of the sample and solvent at the same given temperature, respectively. $I(\theta)^{SSS}_{H2O}$ is the average SHS intensity of water at room temperature. The XX stands for the polarization state of the incident beam relative to the scattering plane (P - parallel or S - perpendicular). Because bulk water is used as normalizing substance all the measured patterns can be compared in their intensity, with an error in the intensity of 1–2 %[65]. The theory and assumptions used to obtain the computed $S(\theta)$ curves in Fig. 1F are detailed in Supplementary Note 1.

## Vibrational sum frequency scattering

The experimental set-up for the vibrational SFS spectroscopy has been described in detail previously[32,43]. Briefly, an 800 nm regeneratively amplified Ti:sapphire system (Spitfire Pro, Spectra physics) seeded with an 80 MHz 800 nm oscillator (Mai Tai SP) was operated at a 1 kHz repetition rate to pump a commercial OPG/OPA/DFG system (HE-TOPAS-C, Light Conversion), used to generate broadband infrared pulses. The visible beam was split off directly from the amplifier, and spectrally shaped with a home-built pulse shaper. The visible (800 nm, 10 mJ, FWHM 15 cm⁻¹) and the IR (3–4.5 mm, 10 mJ, FWHM 170 cm⁻¹) beams were spatially and temporally overlapped with an IR-VIS opening angle of 15°, inside sample cell that had an optical path length of 0.2 mm. At a scattering angle (θ) of 57° as measured in air, the scattered SF light was collimated using a plano-convex lens ($f = 15$ mm, Thorlabs LA1540-B) and passed through two short-wave pass filters (3rd Millenium, 3RD770SP). The SF light was spectrally dispersed with a monochromator (Acton, SpectraPro 2300i) and detected with an intensified CCD camera (Princeton Instruments, PI-Max3) using a gate width of 10 ns. A Glan-Taylor prism (Thorlabs, GT15-B), a half-wave plate (EKSMA, 460-4215) and a polarizing beam splitter cube (CVI, PBS-800-050) and two BaF₂ wire grid polarizers (Thorlabs, WP25H-B) were used to control the polarization of the SF, VIS and infrared beams, respectively. The acquisition time for a single C-H spectrum was 300 s,

while the acquisition time for a single D-O spectrum was 600 s. All measurements were performed in a temperature- and humidity-controlled room ($T = 297$ K; relative humidity, 26.0 %).

The 800 cm⁻¹ spectral range of the O-D stretch modes was obtained by recording different SF spectra spaced by 100 cm⁻¹ frequency steps. The measured sum frequency spectrum at each frequency range was background subtracted and then normalized with IR and VIS pulse energy and acquisition time. The total SF intensity was computed as a weighted sum of the recorded input spectra (Supplementary Note 2), and the resulting intensity was divided by the IR spectral profile, which was obtained by measuring the SF intensity from a thin film of BaTiO₃ nanoparticles (Supplementary Fig. 2).

The effective droplet surface susceptibility ($\Gamma^{(2)}$) spectra, represented by $|\Gamma^{(2)}|^2$, that are shown in Fig. 2B, C, and represent the vibrational surface spectrum of the droplets were obtained by performing a computational procedure that takes into account the extinction of the IR light by absorption, and the efficiency of light collection along the optic axis of the collection optics. The theory behind this procedure was described recently in detail in ref. 66. and its implementation in experiments, together with all the necessary data to do so (frequency-dependent extinction factor, optical response of the focusing geometry) are given in the Supplementary Note 2. The solid lines in Fig. 2B–D are smoothed in Igor Pro 7.0 using moving averages, where each point $x_i$ is replaced by the average of 2 M neighboring values, given by $\bar{x}_i$:

$$\bar{x}_i = \frac{1}{2M+1}\sum_{-M}^{M}x[i+j]$$

## Ab initio molecular dynamics simulations

We used the software package CP2K[67], based on the Born-Oppenheimer approach, to perform ab initio molecular dynamics (AIMD) simulations of hydroxide aqueous solutions and chloride aqueous solutions under the action of static and homogeneous electric fields applied along a given direction (corresponding to the Z-axis). Similar AIMD simulations were performed for aqueous solutions containing neopentane ($C_5H_{12}$) in bulk water. The implementation of external electric fields in numerical codes based on Density Functional Theory (DFT) can be achieved by exploiting the Modern Theory of Polarization and Berry's phases[68-71]. Aqueous samples contained 127 H₂O molecules and one single species of hydroxide and, separately, of chloride (i.e., 383 and 382 atoms, respectively) arranged in cubic cells with side parameter equal to 15.82 Å, so as to reproduce a density of about 1 g cm⁻³. On the other hand, to account for an adequate solvation of neopentane, larger boxes with sides of 19.81 Å and composed of 256 water molecules and one neopentane moiety were employed for the respective simulations (i.e., 785 atoms).

Two electric field regimes were explored during the dynamics of the OH⁻ and Cl⁻ containing aqueous samples, namely the zero-field case and a field intensity of 10⁸ V/m (0.01 V/Å). Albeit the latter value is larger than the typical electrostatic fields used in, e.g., conductivity experiments, such a strength is necessary to explore measurable diffusions of the anions in relatively short AIMD simulations while preventing known field-induced molecular dissociations ( > 0.3 V/Å for the protolysis of water)[72] and vibrational Stark effects[73,74] in the samples. Similarly, a zero-field and two field regimes $0.5 \times 10^9$ and $10^9$ V/m (i.e., 0.05 and 0.1 V/Å) were reproduced in the neopentane AIMD simulations. Dynamics of 100 ps were afforded in the zero-field and 0.1 V/Å cases whilst a shorter trajectory of 20 ps has been produced in the intermediate case for comparison of the molecular dipole distributions. Drifts (Fig. 3D and Supplementary Fig. 6) were determined by discarding the first half of the respective trajectories (i.e., ~50 ps) at zero field and at 0.1 V/Å. Furthermore, a series of self-consistent-field calculations without and in the presence of several electric field

intensities was performed both for the hydrated $OH^-$ and $Cl^-$ species both for the solvated neopentane case (see, e.g., the Hartree potential and the electronic dipole calculations in Fig. 3A, B).

Wavefunctions of the atomic species have been expanded in the TZVP basis set with Goedecker-Teter-Hutter (GTH) pseudopotentials using the GPW method[75]. A plane-wave cutoff of 400 Ry has been imposed. Exchange and correlation (XC) effects were treated with the gradient-corrected Becke-Lee-Yang-Parr (BLYP)[76,77] density functional. Moreover, in order to take into account dispersion interactions, we employed the dispersion-corrected version of BLYP (i.e., BLYP + D3(BJ))[78,79]. A nominal temperature slightly higher than the standard one has been simulated in order to better reproduce the water structure (i.e., $T = 350$ K). The dynamics of ions was simulated classically within a constant number, volume, and temperature (NVT) ensemble, using the Verlet algorithm whereas the canonical sampling has been executed by employing a canonical-sampling-through-velocity-rescaling thermostat (CSVR)[80] set with a time constant equal to 50 fs.

### Many body energy decomposition analysis

The total interaction energy between a single neopentane molecule and the surrounding water solvent at a particular instantaneous geometry is defined as:

$$E_{\text{int}}[NP,wat] = E_{tot}[NP,wat] - E_{tot}[NP] - E_{tot}[wat]$$

where NP and wat correspond to the instantaneous positions of all constituent atomic sites of neopentane and the surrounding waters, respectively, and $E_{tot}[R]$ is the total electronic energy of some molecular (sub)system with fixed nuclear positions, $R$. $E_{\text{int}}$ of the instantaneous configuration of an N-monomer system (here, monomers are taken to be individual NP and water molecules) can be decomposed as a many body expansion (MBE):

$$E_{\text{int}} = E_{2B} + E_{3B} + \ldots + E_{NB}$$

where:

$$E_{2B} = \sum_i E_{2B}[NP,i]$$

is the total 2B interaction energy between NP and the surrounding water as a sum of 2B contributions from the interaction between each water, i, with neopentane, NP, such that:

$$E_{2B}[NP,i] = E_{tot}[NP,i] - E_{tot}[NP] - E_{tot}[i] \tag{4}$$

and

$$E_{3B} = \sum_{i<j} E_{3B}[NP,i,j]$$

with:

$$E_{3B}[NP,i,j] = E_{tot}[NP,i,j] - E_{2B}[NP,i] - E_{2B}[NP,j] - E_{2B}[i,j] - E_{tot}[NP] - E_{tot}[i] - E_{tot}[j]$$

$$= E_{2B}[NP,(i,j)] - E_{2B}[NP,i] - E_{2B}[NP,j] \tag{5}$$

where $E_{2B}[NP,(i,j)]$ is the 2B interaction energy between NP and the water dimer, $(i,j)$, i.e. taking $(i,j)$ to be a single 'monomer' (see Eq. 4):

$$E_{2B}[NP,(i,j)] = E_{tot}[NP,i,j] - E_{tot}[NP] - E_{tot}[i,j]$$

The higher order terms in the MBE similarly can be defined recursively such that the total interaction energy is recovered exactly when including terms up to $E_{NB}$. Each K-order term in the MBE quantifies the collective interaction between K monomers. For example, a given 2B interaction component, $E_{2B}[NP,i]$, quantifies the direct interaction between water i with NP, while a given 3B component, $E_{3B}[NP,i,j]$, quantifies, for example, how the polarization of water i due to j augments the interaction between i and neopentane, etc. It is known that for insulating systems such as water, the MBE expansion converges rapidly, with the 2B and 3B terms accounting for the vast majority of the total interaction energy[81].

The (second generation) absolutely localized molecular orbital energy decomposition analysis (ALMO EDA) method[51–54] decomposes a given interaction energy into the following physically meaningful terms:

$$E_{\text{int}} = E_{elec} + E_{Pauli} + E_{disp} + E_{pol} + E_{ct}$$

where $E_{elec}$ corresponds to the 'permanent' electrostatic energy between the monomers given their unpolarized (isolated monomer) electron densities, $E_{Pauli}$ corresponds to close-range 'Pauli' repulsion due to overlapping monomer densities, $E_{disp}$ corresponds to the dispersion interaction between the monomer densities, $E_{pol}$ corresponds to the relaxation energy due to intermonomer polarization (disallowing charge transfer), and $E_{ct}$ corresponds to the relaxation energy due to intermonomer charge transfer into the fully variational electronic ground state for the particular exchange correlation functional used. ALMO EDA calculations can be further decomposed using the MBE, where, for example, the total 2B ALMO EDA energy terms correspond to pairwise decompositions, computed with Eq. 4, and the total 3B ALMO EDA energy can be computed through the expression on the second line of Eq. 5. All ALMO EDA calculations were done with Q-Chem 5.4[82].

## Data availability

All data in the manuscript and SI are available through Zenodo at https://doi.org/10.5281/zenodo.11532589.

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

## Acknowledgements

S.R. thanks the Julia Jacobi Foundation. S.P. and S.K. acknowledge Swiss National Science Foundation grant 200021-182606-1. A.H. and C.E. acknowledge funding by the European Union (ERC, HyBOP, Grant Number: 101043272). Views and opinions expressed are however those of the author(s) only and do not necessarily reflect those of the European Union or the European Research Council. Neither the European Union nor the granting authority can be held responsible for them. GC acknowledges support from ICSC – Centro Nazionale di Ricerca in High Performance Computing, Big Data and Quantum Computing, funded by the European Union – NextGenerationEU – PNRR, Missione 4 Componente 2 Investimento 1.4. GC is thankful to CINECA for an award under the ISCRA initiative, for the availability of HPC resources.

## Author contributions

S.P., S.K., and T.S. performed the experiments. S.P. and S.R. interpreted the experimental data. G.C. and C.E. performed the MD Simulations and electronic structure calculations. G.C., S.R., A.H., and C.E. analyzed the MD simulations. S.R. conceived and supervised the work. S.R., S.P., G.C., C.E., and A.H. wrote the manuscript.

## Competing interests

The authors declare no competing interests.
