## [Peer Review File · Nature Communications]

nature portfolio

Peer Review File

pH drives electron density fluctuations that enhance electric field-induced liquid flowREVIEWER COMMENTS

Reviewer #1 (Remarks to the Author):

This manuscript challenges the well-established notion that the surface potential (or the surface charge) of a solid-liquid interface is directly related (proportional) to the liquid flow that is induced by an applied electric field tangential to the surface, at least for a specific case of a nanodroplet at pH=11 where the properties of OH⁻ species are important. This is a substantial claim with serious implications, however, unfortunately, for quite a few reasons this story is not convincing from my perspective (see below). Moreover, the manuscript is rather inaccessible, if only because it refers to SI for much more than only "supplementary information", it is basically non-understandable for a non-specialist of the experimental techniques.

But also textually the manuscript generates many comments:

Line 4 of the abstract states that "changing the pH doubles the electrophoresis", this makes no sense without specifying the change of the pH. Line 8 of the abstract states that "classical continuum theory fails under the general case that the pH of the solution is varied", this is just not true for the same reason. These are unjust claims or statements in the abstract.

The very first sentence of the introduction is incorrect: the NS equation describes momentum conservation (and is independent of energy conservation). The condition of an assumed uniform charge distributions (line 8 of introduction) was recently addressed, e.g. by Bonn.

On page 3 it is stated that shifts of the stretch modes of C-H and O-H require a quantum explanation of the charge transfer involving improper H-bonds (without any explanation what this actually is). There is, however, a huge amount of literature that invokes, for instance, the simple concept to Gibbs-energy of transfer that includes all thermodynamics of the transfer from one phase to another. Wouldn't that be a much more conventional interpretation, perhaps together with a classical Donnan potential inside the droplet compared to outside?

Related to this, it remains obscure to me which two models are compared with each other in Figure 3(E). It is quite clear that polarisability is important in this case, also from 2(E), but many classical force fields include polarisability without the need for a quantum calculation (of course a fully quantum calculation includes polarisability), so is (classical) polarisability included in the classical model or not? If not: no wonder that the classical model cannot describe the data; if so, how comparable are the quantum and the (imposed) classical polarisabilities? Also, in 3(E), why do I see only H₂O molecules whereas apparently the OH⁻ is so important?

I also encountered problems with the equations on p.12:

- On line 8, I think polarizability  polarisation? I cannot see that this is consistent with high-pressure results.
- dP/dt is not a charge density fluctuation, the expression for "electrodynamic work" has dimension of power, and the next sentence starts with "This force", it is all just incorrect. On page 12, I am just not convinced by the statement (without reference) that "the force discussed here cannot result in any way from a classical continuum treatment".

As regards the summary:

- My "classical" tendency is strengthened by the statement in the summary that "the polarisation is proportional to the field strength", isn't this just a classical constant polarizability that does not need any quantum explanation? However, the next sentence concerns a "fluctuating electric polarizability"...

Thus, this manuscript leaves me completely confused. It does not provide the information that I would need to properly assess the measurements, and I have the feeling that alternative (and conventional) explanations are not excluded. Clearly, on this basis I cannot recommend this manuscript for publication in Nature Communications.

Reviewer #2 (Remarks to the Author):

The authors investigate the origin of the apparent reduction in zeta potential of neat nanoscale oil droplets when increasing the solution pH. Contrary to the generally accepted notion of a preferential OH⁻ ion adsorption at the oil/water interface -accompanied by an increase in absolute surface charge- the authors propose an alternative perspective. They postulate that the increase in the droplet electrophoretic mobility originates from charge density fluctuations, not accounted for in the classical continuum theory. The key experimental result revolves around the determination (or estimation) of the surface potential at pH 6 and pH 11. using two distinct second-order non-linear optical methods, mainly angle resolved second harmonic scattering (SHS) and vibrational sum frequency scattering (SFS). Surprisingly, the data (Fig 1F and Fig 2C) show that the oil nanodroplets' electrostatic surface potential (and, by extension, also the surface charge) remains unchanged with pH. Ab-initio molecular dynamics give a plausible explanation for the effect. Should these findings withstand the test of time, their implications could be far-reaching in reinterpreting the concept of zeta potential derived from electrophoretic mobilities. Moreover, they could shed light on a spectrum of phenomena involving colloids and electric-field-induced flows. Consequently, the suitability of this work for publication in Nature Communications is apparent.

Nevertheless, it should be noted that the authors may have overstated the generalization of their proposed mechanism, albeit intriguing, given its preliminary nature. Additionally, some discussions tend to magnify or selectively cite examples to unjustly bolster certain interpretations put forward by the authors.

- Introduction. Second paragraph. The pH-induced increase in mobility for poorly wetted surfaces are not "identical", but show instead similar trends (particularly obvious for gas bubbles, which also depend on the chemical nature of the gas. For example: Ushikubo et al. IFAC Proc., Vol 43, Issue 26, 2010, 283-288). The assertion in the last sentence of the second paragraph warrants clarification as it is not strictly true. The surface charging is not identical for all poorly wetted surfaces, and actually depends on the surface chemistry (Kr, Ar, O₂, N₂, air,... bubbles show different mobilities at the same pH and ionic strength).
- The data in Figures 1F and 2C are the experimental centrepieces of the manuscript. However, equivalent sum frequency studies from different research groups (Phys. Rev. Letters 125, 156803 (2020), J. Chem. Phys. 140, 054711 (2014)) at planar alkane/water interfaces show a contrasting outcome. These studies unmistakably show that the water bands and, consequently, the absolute surface potential increase when elevating the pH. Why is it so different at curved interfaces? It would be judicious to reference and briefly delve into the implications of these recent papers in relation to Figure 2C.
- Introduction. Third paragraph. The absence of a clearly discernable peak linked to the OH⁻ in the sum frequency scattering spectra is not proof of its absence at the interface. While the OH⁻ vibrational mode is resolved as a relatively narrow band in Raman spectroscopy for concentrated hydroxide solutions, it is not readily observed in IR. Consequently, the sum frequency "cross-section" for the hydroxide ion ought to be low and further obscured by overlapping OH stretching bands from interfacial water molecules. To my knowledge, there are no reports of a relatively narrow hydroxide anion stretch in the sum frequency literature, even in systems where the OH⁻ is certain to be present at the surface. Maybe the authors can provide a reference?
- The description of the data shown in Figure 1E (pH-dependent surface charge) is misleading. The authors present it as an example of SHS's sensitivity to the surface structure when keeping the ion strength constant (1mM NaCl vs 1mM NaOH). However, like Figure 1D, it is essentially an additional example of the sensitivity of SHS to the surface potential. Increasing the pH deprotonates the silica surface leading to a higher surface charge, thus, surface potential (not surface structure).
- Figure 1C. Is the orange bar really equivalent to -32 mV? In the figure, it appears to be more like -25 mV.
- An explanation of why the fitted lobes in the SHS angle-dependent data (Figure 1F) are much more narrow compared to Figures 1D and 1E, as well as the origin of the asymmetry in the

experimental data in Figure 1F would be instructive. This could be included in the SI following Table S4.

- Page 8. AOT surfactant discussion. The authors neglect to state that the AOT is a negatively charged surfactant and 1mM already corresponds to the CMC. The sentence referring to Figure 2B “Thus, nanodroplet surface adsorption, even at bulk millimolar concentrations, results in drastic changes in vibrational SFS spectra.” is, in this sense, misleading.
- Figure 2D. Charge transfer and improper hydrogen bonds. What is the expected shift in the CH stretching position expected for an increase in the charge? A comparison with a droplet covered with DPPC is not relevant in this context. Given that the spectrometer used had a 15 cm⁻¹ resolution (mentioned in SI), could it resolve a 2 or 3 cm⁻¹ shift?
- How to explain the charging of the gas bubbles? The proposed charge transfer mechanism is understandable for oil droplets, but would the same mechanism be valid for a gas? If not, the overall proposed mechanism should be limited to oil droplets (conclusions).
- Page 10. The current density (J) in A/m² should be conductivity times the electric field (not the molar charge conductivity “sigma” times the electric field “E”).
- While most of the manuscript is generally well written, the abstract is difficult to follow in parts. It should be improved and partly rewritten.

Reviewer #3 (Remarks to the Author):

The paper unfolds a very interesting story on the electrophoretic mobility of oil droplets, which experimentally has been known for a long time to be characterized by a negative zeta potential that quite strongly depends on pH. These findings were controversially discussed in the literature. This paper brings new arguments by comparing SH scattering experiments with ab initio simulations of a hydrophobic molecules in water under the application of an external field. The paper is well written, all the results seem sound, but I have some questions on the argumentations and the interpretations before I can make a definite statement on its suitability for publication.

There is one fundamental point that puzzles me: the electrophoretic mobility is the linear-response to an externally applied electric field, and the mechanism consists of a momentum transfer to the mobile charges that are present at zero electric field in the liquid phase. In this argumentation one does not consider the effect of the electric field on the charge distribution, since this would be a higher-order field effect. Along the same lines, the electric-field-induced dipole moment of an object in the electric field is irrelevant, as dipoles do not move in an electric field, and even if they did, the response would be higher-order in the field strength. In order to get a feeling for the expected zeta potential, the authors should probably calculate the potential on the neopentane, check its sign and its strength, and convert this potential into a zeta potential. Here one has to be careful, since the conversion into a zeta potential requires division by $\epsilon=80$, since the charge-transfer potential is calculated using $\epsilon=1$ while experimental zeta potentials are by convention quoted using $\epsilon=80$ (see Ref 1 for an explanation of this subtle point).

Along the same lines, the argument on page 12 invokes the time derivative of the polarization dP/dt , which is called fluctuation, and which is then inserted into the polarization work. I agree that such time-dependent polarization effects can give rise to transient forces, but the zeta potential measured experimentally is a steady-state effect. If one integrates the polarization force over time, one does not obtain a force that scales linear in time (which would be a steady state force), but rather the difference between the polarization force at the final and initial times, which seems to indicate transient effects (by the way, the theory for electrophoretic effects induced by polarization effects was worked out a few years ago, see Ref 2, so such effects can very well be treated on the continuum level if one accounts for time and spatially dependent polarization fields

and their interaction with time-dependent external fields!)

Ref 1 reported a related ab initio MD study on the electrophoretic mobility of a water slab in a tangential electric field and checked for the expected mobility due to charge transfer effects. The expected electrophoretic mobility based on charge transfer was found to be very small, though. I guess the main difference between the air-water interface and the oil-water interface is that the oil acquires a net negative charge. Quoting that charge of the neopentane molecule more prominently might be useful.

The authors write on page 1 „Curiously, the pH-induced increase in currents¹⁶⁻¹⁹/ mobility²⁰⁻²⁶ is identical in all of these cases, even though the substrates/systems are vastly different.“ But when one compares experimental data from different sources in one plot, as done in fig 1 in Ref 3, one sees that experimental data scatter significantly even for the very same system. This was used in Ref 3 for an impurity mechanism. I know that the current authors advocate a different explanation, but I think they might want to point out that the variations between different experiments are significant.

[1] Electrokinetic, electrochemical, and electrostatic surface potentials of the pristine water liquid–vapor interface, Becker et al, J. Chem. Phys. 157, 240902 (2022), DOI:10.1063/1.5111111

[2] Electrokinetics at Aqueous Interfaces without Mobile Charges, Bonthuis et al, Langmuir 2010, 26(15), 12614–12625

[3] Nanomolar Surface-Active Charged Impurities Account for the Zeta Potential of Hydrophobic Surfaces, Uematsu et al, Langmuir 2020, 36, 3645–3658

Answer to the comments of the referees

We thank all the reviewers for their valuable feedback. In the following, we provide responses to the reviewers' comments. The reviewers' comments are in blue and our responses are in black.

REVIEWER COMMENTS

Reviewer #1 (Remarks to the Author):

This manuscript challenges the well-established notion that the surface potential (or the surface charge) of a solid-liquid interface is directly related (proportional) to the liquid flow that is induced by an applied electric field tangential to the surface, at least for a specific case of a nanodroplet at pH=11 where the properties of OH⁻ species are important. This is a substantial claim with serious implications, however, unfortunately, for quite a few reasons this story is not convincing from my perspective (see below). Moreover, the manuscript is rather inaccessible, if only because it refers to SI for much more than only "supplementary information", it is basically non-understandable for a non-specialist of the experimental techniques.

Author response:

1. We would like to point out that this manuscript deals with a study on **liquid nanodroplet-liquid water interface, not solid-liquid interfaces** as stated by this referee. To avoid any confusion in this regard, the 3rd paragraph of the introduction (page 3) now reads: "Oil nanodroplets in water is a well-studied model system for the hydrophobic/water interface where a pH-dependent increase in electrophoretic mobility has been observed."

On page 4: "We measured electrophoretic flow, electrostatic interface potential and molecular structure as a function of pH via three independent methods on the same hexadecane oil nanodroplet-water system."

2. Understandability of the manuscript to non-specialists

To make the manuscript more accessible to non-specialist readers, we have added a schematic illustration of the origin of the SHS response giving rise to the $\chi_{s,2}^{(2)}$ and Φ_0 contributions in Figure 1B:

On page 7, we have added: " Fig. 1B (right) schematically illustrates the molecular origin of SHS response via $\chi_{s,2}^{(2)}$ and Φ_0 ."

But also textually the manuscript generates many comments:

Line 4 of the abstract states that "changing the pH doubles the electrophoresis", this makes no sense without specifying the change of the pH. Line 8 of the abstract states that "classical continuum theory fails under the general case that the pH of the solution is varied", this is just not true for the same

reason. These are unjust claims or statements in the abstract.

Author response: we have now modified the abstract to clarify these points:

“Liquid flow along a charged interface is commonly described by classical continuum theory, which represents the electric double layer by uniformly distributed point charges. The electrophoretic mobility of hydrophobic nanodroplets in water doubles in magnitude when the pH is varied from neutral to mildly basic (pH 7 - 11). Classical continuum theory predicts that this increase in mobility is due to an increased surface charge. By combining all-optical measurements of surface charge and molecular structure, as well as electronic structure calculations, we show that surface charge and molecular structure at the nanodroplet surface are identical at neutral and mildly basic pH. We propose that the force that propels the droplets originates from two factors: Negative charge on the droplet surface due to charge transfer from and within water, and anisotropic gradients in the fluctuating polarization induced by the electric field. Both charge density fluctuations couple with the external electric field, and lead to droplet flow. Replacing chloride by hydroxide doubles the charge conductivity via the Grotthuss mechanism, and doubles the droplet mobility. This general mechanism deeply impacts a plethora of processes in biology, chemistry, and nanotechnology and provides an explanation of how pH influences hydrodynamic phenomena and the limitations of classical continuum theory currently used to rationalize these effects..”

The very first sentence of the introduction is incorrect: the NS equation describes momentum conservation (and is independent of energy conservation). The condition of an assumed uniform charge distributions (line 8 of introduction) was recently addressed, e.g. by Bonn.

Author response:

We have modified the first sentence as follows: “Liquid flow of an aqueous electrolyte solution in contact with a charged surface is commonly modeled using the incompressible Navier-Stokes equations that combine the continuity equation for the conservation of mass, with momentum conservation, and the Poisson-Boltzmann equation to describe the charge distribution¹.”

In all cases studied in the literature, including those by Bonn, there is the underlying classical assumption that ions are charge carriers, having fixed charges. Since our explanation very clearly excludes the possibility that changes in the ionic charge distribution are the source of the pH-dependent change in mobility, such studies do not contribute to the explanation. In fact, the most relevant studies in this context are not those of Bonn, but Ohshima, and we carefully examined each of these models. Unfortunately, they do not explain our data.

On page 3 it is stated that shifts of the stretch modes of C-H and O-H require a quantum explanation of the charge transfer involving improper H-bonds (without any explanation what this actually is). There is, however, a huge amount of literature that invokes, for instance, the simple concept to Gibbs-energy of transfer that includes all thermodynamics of the transfer from one phase to another. Wouldn't that be a much more conventional interpretation, perhaps together with a classical Donnan potential inside the droplet compared to outside?

Author response: To address the lack of clarity in the part of C-H and O-H mode shifts, we have modified the text on page 3 as follows:

“Recently, an explanation that does not involve ionic groups was proposed³²⁻³⁴: the local topological defects in the hydrogen (H-) bond network of interfacial water creates charge

density oscillations that transfer negative charge density onto the oil surface³⁴. As a result of this interaction, the surface vibrational spectra of interfacial water exhibited a red shift in the O-D stretch modes and a blue shift in the interfacial oil C-H stretch modes. These frequency shifts revealed that charge transfer at the hydrophobic droplet surface involves the formation of improper H-bonds between water and oil³². Thus, water is the key ingredient that enables charge transfer and charge delocalization.”

Donnan potential arises from an unequal distribution of ions between the interface and bulk. Since the SHS response is highly sensitive to the changes in the electrical double layer, if there is a difference in Donnan potential, we would have detected a difference between the SHS response in the presence of 1 mM NaCl and 1 mM NaOH. However, we measured identical second harmonic responses from oil droplets in the presence of 1 mM NaCl and 1 mM NaOH. Therefore, while this explanation offered by the referee might be more conventional, it does not agree with the experimental evidence.

Related to this, it remains obscure to me which two models are compared with each other in Figure 3(E). It is quite clear that polarisability is important in this case, also from 2(E), but many classical force fields include polarisability without the need for a quantum calculation (of course a fully quantum calculation includes polarisability), so is (classical) polarisability included in the classical model or not? If not: no wonder that the classical model cannot describe the data; if so, how comparable are the quantum and the (imposed) classical polarisabilities?

Author response: We have significantly modified the figures and corresponding text associated with the first principles simulations. Besides showing the Hartree potential response of neopentane in the presence of the field and the dipole moment that is induced on the system, we also conduct new calculations that illustrate the importance and the need to go beyond classical induction - all these new results are shown in Figure 3A-C.

We perform a many-body energetic decomposition with a much higher level of electronic structure theory (i.e., ω B97M-V/aug-cc-pVTZ) and demonstrate that the interaction energy between neopentane and water involves a combination of both polarization and charge-transfer contributions. Specifically, the charge-transfer contribution to the interaction energy is approximately 1kBT per water molecule which is indeed significant. Thus, while the reviewer is certainly correct in pointing out that a classical polarizable model may account for some of the phenomena we observe, our new calculations, shown in Figure 3C, prove that, additionally to polarization, charge-transfer effects play a pivotal role. Such a contribution cannot be caught by classical simulation approaches.

Also, in 3(E), why do I see only H₂O molecules whereas apparently the OH⁻ is so important?

From the experimental data presented in Figures 1 and 2, we conclude that OH⁻ ions do not adsorb to the oil droplet surface. Therefore, the pH-induced enhancement in electrophoretic mobility cannot be explained by the surface adsorption of OH⁻ ions. Based on this, we independently investigate the effect of the applied external electric field on the electron density organization around hydrophobic objects as well as bulk OH⁻ and Cl⁻ ions.

Figure 3 shows the effect of an applied electric field on hydrophobic objects in water.

Figure 4 shows the effect of an applied electric field on OH⁻ and Cl⁻ ions in water.

I also encountered problems with the equations on p.12:

- On line 8, I think polarizability  polarisation? I cannot see that this is consistent with high-pressure results.

- dP/dt is not a charge density fluctuation, the expression for "electrodynamic work" has dimension of power, and the next sentence starts with "This force", it is all just incorrect. On page 12, I am just not

convinced by the statement (without reference) that "the force discussed here cannot result in any way from a classical continuum treatment".

Author response:

We clarified the text on page 12 as follows: "To illustrate this point further, we compared the evolution of the dipole moment of hydrated neopentane obtained from the AIMD simulations in the presence of different field strengths. As displayed in Fig. 3B, the electric field significantly polarizes the neopentane molecule in water as witnessed by the progressive growth of the dipole moment from 0.5 to 1.5D. The dipole moment of the water molecules also exhibits subtle changes albeit much smaller compared to that of neopentane. These results point to the important role of electronic polarization enhanced by the external field on simple hydrocarbons in water."

These equations can be found in J. D. Jackson, Classical Electrodynamics Ed. 3.

Regarding the point on high pressure, this was invoked only to make a connection with previous studies that have shown that under pressure, methane can be made soluble in water - this was rooted in an enhancement of its dipole moment (*J. Chem. Phys.* 156, 194504 (2022)).

As regards the summary:

- My "classical" tendency is strengthened by the statement in the summary that "the polarisation is proportional to the field strength", isn't this just a classical constant polarizability that does not need any quantum explanation? However, the next sentence concerns a "fluctuating electric polarizability"...

Author response: We have clarified in our manuscript what is 'classical' and what is 'quantum'. The equations to consider charge and polarization are essentially the same when represented classically or quantum mechanically. The quantum level description that we need in order to explain the data concerns how the charge is described: not as single point charges or charges fixed and localized on atoms, but charges that are not only flexible but also transferable. By explicitly describing charge distributions on the quantum level, we observe charge transfer across the interface and within the 3D hydrogen bond network of water. It is these elements that are needed to explain the data (as well as other experiments in the literature).

Experimental evidence in this manuscript is as follows:

- Increasing the pH at constant ionic strength does not lead to a change in the electrostatic surface potential of nanodroplets dispersed in water when directly measured in the absence of an electrostatic field.
- The interfacial structure of oil droplet (C-H modes) and water (O-D modes) is pH-independent. Both spectral shapes as well as the measured intensities are completely pH-independent.

Previous experiments (*Science* **374**,1366-1370(2021).DOI: [10.1126/science.abj3007](https://doi.org/10.1126/science.abj3007)), *J. Phys. Chem. B* **2022**, **126**, **16**, **3186–3192**. <https://pubs.acs.org/doi/10.1021/acs.jpcc.2c01987> , *Chem. Phys. Lett.* **615**, 124-131, 2014. <https://doi.org/10.1016/j.cplett.2014.09.034>.) showed that:

- For negatively charged pure oil droplets in water the sum frequency vibrational spectrum of interfacial water displays up to 150 cm⁻¹ frequency shifts towards lower frequencies. These shifts emerge from 2 independent sets of experiments, one employing polarimetry, the other isotope dilution.

- For negatively charged pure oil droplets in water, the C-H modes of the oil are blue shifted by 5 cm^{-1} . Inverting the charge at the interface or shielding the oil phase from the water by means of a densely packed lipid monolayer removes these shifts.
- Interference effects between the vibrational peaks of C-H groups at the oil droplet interface and the O-H groups from water are pH-independent

Other observations of note are:

- The pH-dependent change in conductivity of an aqueous solution is the only physical parameter that relates to the physics at hand and is very strongly pH-dependent.
- Numerous pH-dependent electrophoretic/electrokinetic phenomena show a remarkably similar pH-dependence:
 - at concentrations far below the effect of other simple ions [introduction]
 - at interfaces that are chemically diverse, spanning the liquid and solid phase and many chemically different substrates

For these reasons, an explanation that involves changes in the classical charge distribution in the electric double layer does not agree with the experimental evidence.

We have re-written the summary as follows:

“Remarkably, numerous different experiments involving liquid flow along charged surfaces on display very similar pH dependent effects, despite their surface chemistries being wildly different. Here, we provided a general framework for rationalizing the origins of these phenomena. A classical continuum model description of electrophoretic phenomena breaks down under the very general case when: (i) Interfacial charge transfer effects become important, i.e. on poorly wetted surfaces in water, and (ii) the pH of the solution is varied, involving the Grotthuss mechanism. This warrants a deeper investigation into the electro-hydrodynamics of hydrophobic objects in water as a function of pH. We therefore measured electrophoretic mobility of hydrophobic nanoscale oil droplets in water and combined it with nonlinear optical spectroscopic measurements as a function of pH. Surface potential and charge was measured as a function of pH under conditions of constant ionic strength whereby Cl^- and OH^- were interchanged. We also used vibrational sum frequency scattering to determine the pH-dependent molecular interfacial structure and interactions. While the mobility increased drastically upon increasing the pH, neither did the surface potential, nor the interfacial interactions. In other words, the surface charge turned out to be pH-independent.

Ab initio molecular dynamics (AIMD) simulations of neopentane in water (“the smallest oil droplet”) with an external electric field show an electronic charge polarization within the molecule, which is proportional to the external field strength. Applying an electrostatic field across solutions of OH^- and Cl^- ions using AIMD leads to a different polarization, with an electron density distribution that is both enhanced and anisotropic for OH^- , but not for Cl^- . Besides the translational drift of the neopentane in the presence of field, visual inspection of

our simulations also indicates that the molecule also undergoes rotational motion. This effect is likely due to the anisotropic response of charge in the presence of the field. The coupling of these translational and rotational motion is another aspect that warrants further study.

Combining experimental findings and quantum-level considerations, we arrive at a description of an electric-field-induced net force that originates from gradients in the fluctuating electric polarizability of the droplet surface, which ultimately arises from electron density fluctuations leading to charge transfer. Although it is well known that charge gradients in external fields – or field gradients with fixed charge distributions – give rise to forces, as e.g. manifested in dielectrophoresis⁵⁶ or electrorotation⁵⁷, the force discussed here cannot result in any way from a purely classical continuum treatment, as the charges in such a description are not permitted to be delocalized across different molecules, and would therefore not display the type of behavior that one finds using a quantum-mechanical approach. At the same time, our experiments clearly indicate that the surface charge and ion distribution in the electric double layer is not changing as a function of pH. The important role of electronic polarization as well as that of the Grotthuss mechanism, indicates that one must go beyond classical continuum models.

The induced charge density gradient interacts with the external electric field itself and results in a force, as it represents a gradient in the electrodynamic energy distribution. This force is a more complete rendering of $F=qE$, considering not only classical charges but also dynamic fluctuations of charge density displacements. On neopentane, the field that acts on the induced dynamic polarization induces a mobility of the same order of magnitude as that experimentally measured in the nanodroplets. In basic solutions, the global charge asymmetry / polarization gradient is enhanced by $\sim x 2$ because the charge conductivity (and thus current density / change in polarizability) is larger by exactly the same amount, and it couples to the electrodynamic energy. In the case of gas-bubbles, the charge transfer mechanism involves only the water molecules close to the surface which leads to an effective negative charge. This mechanism is very general and impacts many processes in biology, chemistry, and nanotechnology and provides an explanation of why bulk acidity or basicity can have important implications in transport phenomena.”

Thus, this manuscript leaves me completely confused. It does not provide the information that I would need to properly assess the measurements, and I have the feeling that alternative (and conventional) explanations are not excluded. Clearly, on this basis I cannot recommend this manuscript for publication in Nature Communications.

Author response: We believe that the additional data from simulations in combination with the modifications to the text have made the manuscript clearer.

Reviewer #2 (Remarks to the Author):

The authors investigate the origin of the apparent reduction in zeta potential of neat nanoscale oil droplets when increasing the solution pH. Contrary to the generally accepted notion of a preferential OH⁻ ion adsorption at the oil/water interface -accompanied by an increase in absolute surface charge- the authors propose an alternative perspective. They postulate that the increase in the droplet electrophoretic mobility originates from charge density fluctuations, not accounted for in the classical continuum theory. The key experimental result revolves around the determination (or estimation) of the surface potential at pH 6 and pH 11. using two distinct second-order non-linear optical methods, mainly angle resolved second harmonic scattering (SHS) and vibrational sum frequency scattering (SFS). Surprisingly, the data (Fig 1F and Fig 2C) show that the oil nanodroplets' electrostatic surface potential (and, by extension, also the surface charge) remains unchanged with pH. Ab-initio molecular dynamics give a plausible explanation for the effect.

Should these findings withstand the test of time, their implications could be far-reaching in reinterpreting the concept of zeta potential derived from electrophoretic mobilities. Moreover, they could shed light on a spectrum of phenomena involving colloids and electric-field-induced flows. Consequently, the suitability of this work for publication in Nature Communications is apparent.

Nevertheless, it should be noted that the authors may have overstated the generalization of their proposed mechanism, albeit intriguing, given its preliminary nature. Additionally, some discussions tend to magnify or selectively cite examples to unjustly bolster certain interpretations put forward by the authors.

Author response: We thank the referee for this assessment. In addition to the present work, we now have further experimental evidence of the importance of charge transfer on very different systems as well. While these are not published they do increase our confidence that the present interpretation is correct. We have now also added a significant amount of discussion into the paper both on the experimental and theoretical side that provides more context as well as highlights the aspects of the results in a more transparent and fair manner.

• Introduction. Second paragraph. The pH-induced increase in mobility for poorly wetted surfaces are not "identical", but show instead similar trends (particularly obvious for gas bubbles, which also depend on the chemical nature of the gas. For example: Ushikubo et al. IFAC Proc., Vol 43, Issue 26, 2010, 283-288). The assertion in the last sentence of the second paragraph warrants clarification as it is not strictly true. The surface charging is not identical for all poorly wetted surfaces, and actually depends on the surface chemistry (Kr, Ar, O₂, N₂, air,... bubbles show different mobilities at the same pH and ionic strength).

Author response: The magnitude of the pH-induced increase in mobility is similar, not the absolute value of the mobility itself.

To clarify this, we have modified this paragraph as follows:

"As examples of this challenge, the following well-known pH-induced phenomena can be considered: Pressure-driven electro-osmosis through hexagonal boron nitride nanocapillaries¹⁵, carbon nanotubes¹⁶, graphene oxide membrane conductance¹⁷, and streaming currents through MoS₂ pores¹⁸, all of which have vastly different surface chemistries in aqueous solutions. All these diverse experiments display a remarkable increase in current (x 2-3) when the pH of the surrounding bulk solution is increased from neutral to mildly basic (1 mM NaOH, pH 11). Hydrophobic nanoparticles, droplets, and gas bubbles of different materials all display the same pH-dependent change in electrophoretic mobility¹⁹⁻²⁵. Increasing the pH of the surrounding aqueous bulk phase to the same mildly basic value results also in a x 2-3 increase in droplet mobility under the influence of an external electrostatic field. The involved poorly wetted surfaces in the above examples are negatively charged. According to classical-continuum theory, this charge more than doubles under the influence of small changes in the bulk concentration of OH⁻, increasing the current /

mobility in an external electrostatic field^{19,26} (due to the well-known electrostatic force of external field \mathbf{E} on a charge q , $\mathbf{F}=q\mathbf{E}$). Curiously, the pH-induced increase in currents¹⁵⁻¹⁸/mobility¹⁹⁻²⁵ is similar in all of these cases, even though the substrates/systems are vastly different. It is well-known that ion-interface interactions are very sensitive to the ionic species and the chemical composition of the interface, but such effects manifest themselves > 0.1 M ionic strength²⁷, i.e. well above mildly basic conditions (<1 mM). Therefore, from a surface chemistry perspective, it is highly unlikely that dilute hydroxide bulk concentrations should lead to identical surface charging due to the surface activity of the hydroxide ions in this wide variety of systems.”

• The data in Figures 1F and 2C are the experimental centrepieces of the manuscript. However, equivalent sum frequency studies from different research groups (Phys. Rev. Letters 125, 156803 (2020), J. Chem. Phys. 140, 054711 (2014)) at planar alkane/water interfaces show a contrasting outcome. These studies unmistakably show that the water bands and, consequently, the absolute surface potential increase when elevating the pH. Why is it so different at curved interfaces? It would be judicious to reference and briefly delve into the implications of these recent papers in relation to Figure 2C.

The studies by Bakker (2014) and Tian (2020) on planar oil/water interfaces show an increase in the O-H or O-D spectral region of the planar vibrational SFG-water interface. In the absence of a clear vibrational signature of hydroxyl ions (IR and Raman active and thus SFG visible; see next answer), the change is attributed to the adsorption of OH⁻. Within this framework/assumption, Tian et al, use the Gouy-Chapman model to compute the surface potential. In this analysis, the number of variables is larger than the number of computed values. Neither study uses a constant ionic strength, which can also explain the observation. Also, there is no streaming potential measurement on these interfaces. Therefore, we conclude that the presented data itself does not disagree with our data. The chosen interpretation disagrees, though.

In our study we do have all observables: O-D spectra, C-H spectra, AR-SHS data, and electrophoretic mobility. The AR-SHS experiment uses 2 independent data sets that are describable with 2 unknowns, one of which is the surface potential. There is also no electric-double layer model needed to get to a surface potential value, which means it is more robust than the Tian / Bakker study.

We have included a section in the SI that deals with these and other questions about the literature, and it is pasted here below:

“S1. Relation to previous experiments and considerations around impurities

Historically, the pH-dependent increase in droplet mobility has been explained in terms of an increase in charge density. Because the droplet speed in an electrostatic field depends linearly on the adsorbed surface charge, this would explain the mobility increase. An increase in charge density necessarily also results in an increase in the surface potential. For low surface charge densities and at constant ionic strength, $s_0 \sim F_0$. Therefore, if $s_0(\text{pH } 11) = 2.2 \times s_0(\text{pH } 7)$, then $F_0(\text{pH } 11) = 2.2 \times F_0(\text{pH } 7)$. In Fig. 2, we directly measure F_0 , and this measurement does not require any input on how F_0 relates to s_0 . The measured intensity scales with F_0^2 , and thus, we expect at least an increase in the intensity by a factor of ~ 4 , which is indeed what the data in Fig. 1D (green data, $F_0 = -60$ mV; blue data $F_0 = -109$ mV) and the model of Fig. 1F (dashed/solid lines) provide. Therefore, while the increase in droplet mobility agrees with literature, and could in the absence of any other experiments be explained by a pH-dependent increase in s_0 , it is not reconcilable with the AR-SHS measurements of Fig. 1F. In addition, the SFS measurements are also sensitive to F_0 , in a very similar manner as the AR-SHS measurements, and an increase in F_0 , would have been

clearly visible as a change in the O-D stretch intensity. In fact, this is seen in Fig. 2B, where the SFS O-D spectrum of 1 vol% 200 nm radius hexadecane droplets in a 1 mM AOT-water solution is plotted. Here, the O-D response has increased drastically compared to that of neat ultrapure oil droplets in water. Similar vibrational SFS experiments have been conducted in the past decades in Refs^{33, 54}. Various pH-dependent aspects were reported, always in combination with ζ -potential measurements: The phase between oil and water responses⁵⁴, and the weakly resonant D-O tail of vibrational SF response³³. Both these metrics should be changing if the surface charge changes. While the z-potential was pH dependent and reproduced literature values, neither the oil-water phase⁵⁴ nor the weakly resonant SF D-O response showed any sign of pH dependence in the pH range 7-12³³, which is consistent with the present observation and explanation.

Next, we briefly comment on the aspect of impurities. z-potentials as measured at different pH values are in the range [30 -120] mV¹⁹. Applying the Poisson-Boltzmann equation¹, this corresponds to surface charge densities in the range ~ 0.01 (100) – 0.3 (3.33) e/nm² (nm²/e⁻). Since these are low values, several studies, some theoretical^{30, 55}, have brought forth the possibility that impurities are responsible for the pH dependence in the z-potential. These impurities are hypothesized to arise from the oil (in the form of carboxylates²⁹), the water (in the form of an undefined agent with a specific pK_a³⁰, or atmospheric CO₂²⁸, or from what appears to be the walls of the glass cuvette when it is subjected to > 24h of sonication^{41, 56}. In neither of these studies independent surface potential measurements were conducted, and direct molecular level evidence was absent.

Impurities are unwanted substances that derive from local geographic conditions and are indeed an inescapable element in any experiment with chemicals. In most scientific domains experimental reproducibility of data is taken as evidence for the relative unimportance of impurities, since they tend to vary in time and space. Here, the pH-dependent change in electrophoretic effects is seen in diverse experiments¹⁵⁻²⁵, conducted across different eras (1930's – 2020's), and on different continents (Europe, Oceania, America, Asia). Thus, it would be highly unlikely that impurities are at the origin of the increase in mobility or conductivity. Experiments that do show a lot of variability are reflection SFG experiments⁵⁷⁻⁶² conducted on planar oil-water interfaces, as discussed in detail in Ref.³² (section S1). Various groups have aimed to measure the water/oil molecular structure and reported vastly different SFG spectra, even a few years apart within the same laboratory. Due to the difficulty of making a pristine planar oil/water interface in combination with the very small surface to volume ratio, we expect that in this case, impurities might play a role. Also, in relation to answering the question at hand in this work, direct surface potential and streaming potential measurements have not been performed together with the SFG studies, which makes it impossible to correlate electrophoresis to electrostatic potential and molecular structure. In contrast to extended planar oil-water interfaces, oil nanodroplets in water are easy to prepare reproducibly, have a $\sim 10^4$ x higher surface to volume ratio and allow for measuring electrophoresis, electrostatic potential and molecular structure on the same sample. Based on the above discussion (reproducible data, high surface to volume ratio) we expect the influence of impurities to be unimportant in this case.

Nevertheless, we have in the past published several studies in which we quantified the role of impurities in our experiments (see Refs.^{31, 41, 63, 64}): We examined the effect of impure oil³¹, the possibility of impure water and salt, the effect of dissolved CO₂⁶³, as well as impurities leaching from glass wall containers⁴¹. We also measured the level of impurity in the chemicals used in our experiments, and determined the sensitivity of our experimental tools: The concentration of impurities in our samples is $\sim < 1$ nM, primarily a consequence of the sensitivity of the methods to determine these⁶³. Furthermore, we determined the surface sensitivity of our techniques on 100 nm radius objects in a 1 vol % dispersion (SFS), or a 0.05 vol % dispersion (SHS). We previously³¹ determined that vibrational sum frequency scattering has a sensitivity of detecting 1 molecule / 27 nm², which is presently improvable by

a factor of ~ 5 by varying detection settings⁶⁵. For AR-SHS the sensitivity is better: AR-SHS can detect a single molecule adsorbing on a ~ 100 nm object, i.e. 1 molecule / 10^5 nm² (in the femtomolar range in terms of bulk concentration⁶⁶. Equivalent non-resonant SH imaging detects a single oriented water molecule in a ~ 320 nm³ bulk volume⁶⁷, which converts to a similar sensitivity value for a droplet interface. Therefore, we conclude that on the basis of reproducibility of results (in our lab over a period of almost 20 years), samples with surface to volume ratios $\sim 10^4$ x higher than in other surface sensitive measurements, and the excellent (SFS) and exquisite interfacial sensitivity (SHS) the role of impurities in this context is negligible.”

• Introduction. Third paragraph. The absence of a clearly discernable peak linked to the OH⁻ in the sum frequency scattering spectra is not proof of its absence at the interface. While the OH⁻ vibrational mode is resolved as a relatively narrow band in Raman spectroscopy for concentrated hydroxide solutions, it is not readily observed in IR. Consequently, the sum frequency "cross-section" for the hydroxide ion ought to be low and further obscured by overlapping OH stretching bands from interfacial water molecules. To my knowledge, there are no reports of a relatively narrow hydroxide anion stretch in the sum frequency literature, even in systems where the OH⁻ is certain to be present at the surface. Maybe the authors can provide a reference?

The IR and Raman cross-sections and peak shapes are in fact equally large, this can be found e.g. in Ref. [Hermannson et al, Chemical Physics Letters 514 (2011) 1–15], and was also reproduced experimentally in our lab. There are SFG spectra of CaF₂ – water surfaces whereby the F⁻ ions are replaced by OH⁻ ions at high pH (Khatib, R. et al. Water orientation and hydrogen-bond structure at the fluorite/water interface. Sci. Rep. **6**, 24287; doi: 10.1038/srep24287 (2016).)). This replacement gives rise to a narrow peak at ~ 3645 cm⁻¹, precisely where OH⁻ is indeed expected. On these grounds, it would also have to appear at the oil-water interface if it is there.

• The description of the data shown in Figure 1E (pH-dependent surface charge) is misleading. The authors present it as an example of SHS's sensitivity to the surface structure when keeping the ion strength constant (1mM NaCl vs 1mM NaOH). However, like Figure 1D, it is essentially an additional example of the sensitivity of SHS to the surface potential. Increasing the pH deprotonates the silica surface leading to a higher surface charge, thus, surface potential (not surface structure).

Fig. 1D shows the effect of doubling the surface potential. Fig. 1E shows the effect of replacing 1 mM NaCl by 1 mM NaOH. In the latter case, both the surface susceptibility as well as the surface potential change.

Modification to the manuscript (p. 7):

We changed: “Both these examples demonstrate the unique sensitivity of the SHS experiment to surface potential (1D) and surface structure (1E).”

To “Both these examples demonstrate the unique sensitivity of the SHS experiment to surface potential (1D) and surface structure and surface potential (1E).”

We also included the following statement in the SI section S1:

Furthermore, we determined the surface sensitivity of our techniques on 100 nm radius objects in a 1 vol % dispersion (SFS), or a 0.05 vol % dispersion (SHS). We previously³¹ determined that vibrational sum frequency scattering has a sensitivity of detecting 1 molecule / 27 nm², which is presently improvable by a factor of ~ 5 by varying detection settings⁶⁵. For AR-SHS the sensitivity is better: AR-SHS can detect a single molecule adsorbing on a ~ 100 nm object, i.e. 1 molecule / 10^5 nm² (in the femtomolar range in terms of bulk concentration⁶⁶. Equivalent non-resonant SH imaging detects a single oriented water molecule in a ~ 320 nm³ bulk volume⁶⁷, which converts to a similar sensitivity value for a droplet interface. Therefore,

we conclude that on the basis of reproducibility of results (in our lab over a period of almost 20 years), samples with surface to volume ratios $\sim 10^4$ x higher than in other surface sensitive measurements, and the excellent (SFS) and exquisite interfacial sensitivity (SHS) the role of impurities in this context is negligible.

- Figure 1C. Is the orange bar really equivalent to -32 mV? In the figure, it appears to be more like -25 mV.

This has now been corrected in the updated figure 1.

- An explanation of why the fitted lobes in the SHS angle-dependent data (Figure 1F) are much more narrow compared to Figures 1D and 1E, as well as the origin of the asymmetry in the experimental data in Figure 1F would be instructive. This could be included in the SI following Table S4.

We have included the following text in the SI after table S4: “From Fig. 1F, the simulated SHS curve is narrower than the experimental SHS patterns. The difference between the width of the fits is likely caused by a somewhat broad size dispersion of this type of droplets that was not included in the computation.”

- Page 8. AOT surfactant discussion. The authors neglect to state that the AOT is a negatively charged surfactant and 1mM already corresponds to the CMC. The sentence referring to Figure 2B “Thus, nanodroplet surface adsorption, even at bulk millimolar concentrations, results in drastic changes in vibrational SFS spectra.” is, in this sense, misleading.

In contrast to what is commonly observed at planar interfaces, negatively charged surfactants do not form densely packed layers at oil droplet water interfaces. This observation is a somewhat logical outcome of the balance of interactions which is different on the sub-micromolar length scale, see: ACS Nano 2017, 11, 12111–12120. Furthermore, we have now added explicit detection sensitivities as mentioned in answer 4.

- Figure 2D. Charge transfer and improper hydrogen bonds. What is the expected shift in the CH stretching position expected for an increase in the charge? A comparison with a droplet covered with DPPC is not relevant in this context. Given that the spectrometer used had a 15 cm^{-1} resolution (mentioned in SI), could it resolve a 2 or 3 cm^{-1} shift?

The shifts are $\sim 5 \text{ cm}^{-1}$ in magnitude and are visible as shifts of the center frequency, even when the resolution is greater than 5 cm^{-1} . However, we have also previously performed measurements at better resolution ($\sim 3 \text{ cm}^{-1}$). The shifts remain the same. It is instructive to think of the relation between ‘center frequency’ and ‘resolution’ in terms of single molecule microscopy: while the optical resolution is determined by the diffraction limit, appropriate analysis can achieve localization accuracies of much shorter distances.

- How to explain the charging of the gas bubbles? The proposed charge transfer mechanism is understandable for oil droplets, but would the same mechanism be valid for a gas? If not, the overall proposed mechanism should be limited to oil droplets (conclusions).

Based on previous AIMD simulations, for gas bubbles there will also be a charge separation, but here the charge transfer is only within the aqueous phase due to the accumulation of coordination defects [Nature Communications 11, (901) 2020.]. The pH mechanism is purely dependent on the bulk aqueous phase, and thus we expect that the role of the bulk polarizability of the solution to be very similar. We have added to the conclusions some sentences clarifying this point.

- Page 10. The current density (J) in A/m² should be conductivity times the electric field (not the molar charge conductivity “sigma” times the electric field “E”).

We have updated the text. It now reads: “ As a consequence of this difference, dilute OH⁻ solutions conduct charge $\times \sim 2$ faster than dilute Cl⁻ solutions: The molar conductivity, Λ_m , is 244.60 S cm²/mol for 1 mM NaOH in water and 123.68 S cm²/mol for 1 mM NaCl⁵¹, even though the dielectric constant and viscosity of both solutions are identical. Given a certain applied electrostatic field (E) to a solution with a conductivity σ , the current density $J = \sigma \cdot E$ is therefore $\times \sim 2$ bigger, which means that the solutions' polarization P must also be bigger (since they are related via $j = dP/dt$).

- While most of the manuscript is generally well written, the abstract is difficult to follow in parts. It should be improved and partly rewritten.

With the aim of improving its readability, the abstract has completely been rewritten.

Reviewer #3 (Remarks to the Author):

The paper unfolds a very interesting story on the electrophoretic mobility of oil droplets, which experimentally has been known for a long time to be characterized by a negative zeta potential that quite strongly depends on pH. These findings were controversially discussed in the literature. This paper brings new arguments by comparing SH scattering experiments with ab initio simulations of a hydrophobic molecules in water under the application of an external field. The paper is well written, all the results seem sound, but I have some questions on the argumentations and the interpretations before I can make a definite statement on its suitability for publication.

There is one fundamental point that puzzles me: the electrophoretic mobility is the linear-response to an externally applied electric field, and the mechanism consists of a momentum transfer to the mobile charges that are present at zero electric field in the liquid phase. In this argumentation one does not consider the effect of the electric field on the charge distribution, since this would be a higher-order field effect. Along the same lines, the electric-field-induced dipole moment of an object in the electric field is irrelevant, as dipoles do not move in an electric field, and even if they did, the response would be higher-order in the field strength. In order to get a feeling for the expected zeta potential, the authors should probably calculate the potential on the neopentane, check its sign and its strength, and convert this potential into a zeta potential. Here one has to be careful, since the conversion into a zeta potential requires division by $\epsilon=80$, since the charge-transfer potential is calculated using $\epsilon=1$ while experimental zeta potentials are by convention quoted using $\epsilon=80$ (see Ref 1 for an explanation of this subtle point).

This is an interesting point. Experimentally we have measured by now a number of systems where we compared the surface potential as measured by SHS to the zeta potential derived from electrokinetic mobility. What emerges from this comparison is that surface potentials are generally consistent with expected changes in the electric double layer/surface charge distributions while the zeta potentials are not [J. Phys. Chem. C 2019, 123, 33, 20393–20404; J. Phys. Chem. C 2020, 124, 204, 10961–1097]. The most common go-to argument to explain this difference is the change of the position of the slip plane, which alters zeta potentials but not surface potentials. Interestingly, it cannot be measured/verified whether this is the case.

We now report in the manuscript the magnitude of the charge of neopentane in the manuscript. In addition, from the mobility, we also extract an effective zeta-potential which is now reported in the paper and is in modest agreement with that of the experiments.

Along the same lines, the argument on page 12 invokes the time derivative of the polarization dP/dt , which is called fluctuation, and which is then inserted into the polarization work. I agree that such time-dependent polarization effects can give rise to transient forces, but the zeta potential measured experimentally is a steady-state effect. If one integrates the polarization force over time, one does not

obtain a force that scales linear in time (which would be a steady state force), but rather the difference between the polarization force at the final and initial times, which seems to indicate transient effects (by the way, the theory for electrophoretic effects induced by polarization effects was worked out a few years ago, see Ref 2, so such effects can very well be treated on the continuum level if one accounts for time and spatially dependent polarization fields and their interaction with time-dependent external fields!)

Our current interpretation of the origin of the pH dependence on the mobility of the droplet is that it arises from two effects: 1) The intrinsic field-independent charge transfer that results in a net charge on the oil-droplet and 2) A field induced force due to the polarizability. The latter effect results in a measurable current and is therefore not a transient effect. We admit that the microscopic mechanism by which the enhanced polarizability due to the Grotthuss mechanism, remains an open question. We have made a serious attempt in the revision of the manuscript to present what is known as well as what is not. However, we believe that this paper should motivate and trigger important discussions and work on the problem.

Ref 1 reported a related ab initio MD study on the electrophoretic mobility of a water slab in a tangential electric field and checked for the expected mobility due to charge transfer effects. The expected electrophoretic mobility based on charge transfer was found to be very small, though. I guess the main difference between the air-water interface and the oil-water interface is that the oil acquires a net negative charge. Quoting that charge of the neopentane molecule more prominently might be useful.

Author response: We have included this in the manuscript in the caption of Fig 2: "The surface charge density in the oil phase in contact with pH neutral water as arising from charge-transfer effects was computed to be $\sim -0.015 \text{ e/nm}^2$, Ref.³⁴"

We also included a paragraph on p.13 discussing expected charge densities:

"Indeed, the neopentane drifts in the opposite direction of the field (while it does not do so when the field is off). This is consistent with the neopentane acquiring an effective net negative charge of -0.05 e^- (using a Mulliken charge scheme) from the interaction with water.

As alluded to earlier, the field strengths used in the simulations are rather large in magnitude (0.1 V/\AA) in order to capture the observed flow on realistic timescales. Nonetheless, our simulations yield a mobility $-1 \cdot 10^{-8} \text{ m}^2/\text{Vs}$ and a corresponding ζ -potential of $\sim -30 \text{ mV}$ of the neopentane molecule. Both these numbers are in fairly good agreement with the experiment (Fig. 1C) despite the theoretical and computational limitations."

We also note that, the slip length could be a lot bigger than what is used, e.g. $> 10 \text{ nm}$, this will have a drastic impact on the computed mobility (Scientific Reports, (2019) 9:18957, <https://doi.org/10.1038/s41598-019-55491-2>).

The authors write on page 1 „Curiously, the pH-induced increase in currents^{16-19/} mobility²⁰⁻²⁶ is identical in all of these cases, even though the substrates/systems are vastly different.“ But when one compares experimental data from different sources in one plot, as done in fig 1 in Ref 3, one sees that experimental data scatter significantly even for the very same system. This was used in Ref 3 for an impurity mechanism. I know that the current authors advocate a different explanation, but I think they might want to point out that the variations between different experiments are significant.

[1] Electrokinetic, electrochemical, and electrostatic surface potentials of the pristine water liquid–vapor interface, Becker et al, J. Chem. Phys. 157, 240902 (2022);

[2] Electrokinetics at Aqueous Interfaces without Mobile Charges, Bonthuis et al, Langmuir 2010, 26(15), 12614–12625

[3] Nanomolar Surface-Active Charged Impurities Account for the Zeta Potential of Hydrophobic Surfaces, Uematsu et al, Langmuir 2020, 36, 3645–3658

Author response: Indeed, there are experimental fluctuations, and some comparisons were harder to make than others due to the lack of data. The magnitude of the electrophoretic mobilities themselves are different between systems, but the change going from neutral pH to pH 11 is similar.

We have changed the sentence on page 1: “Curiously, the pH-induced increase in currents¹⁵⁻¹⁸/ mobility¹⁹⁻²⁵ is very similar in all of these cases, even though the substrates/systems are vastly different.”

REVIEWER COMMENTS

Reviewer #2 (Remarks to the Author):

The authors have significantly improved the manuscript in their revised version, both in terms of clarity and argumentation. The experimental observation that the oil nanodroplets' surface potential (& surface charge) remains constant with increasing pH is indeed puzzling and unexpected. The proposed mechanism, in terms of fluctuating polarizability gradients to explain the observed differences in electrophoretic mobilities with pH, appears to be plausible. I recommend this manuscript for publication in Nature Communications, subject to the additional minor comments below.

a) In the newly added text in the manuscript, it is repeatedly stated that the charge conductivity in the bulk aqueous solution increases 2.2x when exchanging Cl⁻ to OH⁻ (see, for instance, page 4, introduction, and page 16, paragraph before Summary). However, the increase in conductivity should be just 2x for NaCl compared to NaOH, and 2.6x if only considering the anions. Including a reference will be valuable. I note, however, that on page 14 (polarization and the Grotthuss mechanism), the authors mentioned a value of x2 instead. A related comment: On page 4, in the last paragraph, it is stated that the polarizability of the bulk solution also increases by x2.2. Is it really so?

b) In connection to the authors' rebuttal to the second comment of Reviewer#2, they state that the data presented in Figures 1F and 2C, "does not disagree" with previously published by Bakker (2014) and Tian (2020) on flat oil/water interfaces using a similar, yet more established, non-linear optical technique. This is not accurate. The previous studies show, without doubt, the opposite behaviour, clearly indicating that the surface charge increases (becomes more negative) at mildly basic conditions. The authors argue that previous studies did not keep the ionic strength constant. However, at least in the Phys. Rev. Letters 125, 156803 (2020) study, the influence of the ionic strength was explicitly considered in the main text, and particularly in their SI. The authors also appear to suggest potential contaminants could be the cause for the disagreement, but Tian (2020) used submonolayer amounts of purified oil (hexane), representing surface-to-volume ratios similar, if not higher, than those for the nanodroplets, indicating that contamination is unlikely (at least not more than in the nanodroplets). I want to stress that experimental evidence shows that the surface is more negatively charged at flat oil/mildly basic aqueous interfaces, which must be mentioned in the main text. Curved interfaces can behave differently, which has been previously proven by others, including some of the authors in the same manuscript. Thus, it should have no consequences in the manuscript discussions but limit its scope.

c) Authors' rebuttal to the third comment from Reviewer#2 (a lack of a clearly discernible peak linked to the OH⁻ in the sum frequency scattering is not proof of its absence at the interface because, to my knowledge, it has never been reported). The authors provide a reference for CaF₂-water surfaces (Khalib et al., though first reported by Becraft and Richmond, Langmuir 2001, 17, 25, 7721). However, in those studies, the "narrow peak at 3645 cm⁻¹" originates from the CaOH, not the OH⁻ anion. Since the authors cannot find a suitable reference, my fears appear justified: the lack of a discernible peak in the sum frequency scattering spectra is not proof of its absence at the interface. The concerned text should be edited accordingly.

Reviewer #3 (Remarks to the Author):

The authors have adequately addressed my comments. As they say in their reply:

"We have made a serious attempt in the revision of the manuscript to present what is known as well as what is not. However, we believe that this paper should motivate and trigger important discussions and work on the problem"

Not everything is fully understood on a molecular level, but the paper presents significant progress on a very important and highly debated topic in the physical chemistry of interfaces, so I support publication.

We thank both referees for reviewing our manuscript. Below, we provide our responses to the reviewer's questions. The reviewer's question is given in blue, and our response is given in black.

Reviewer #2 (Remarks to the Author):

The authors have significantly improved the manuscript in their revised version, both in terms of clarity and argumentation. The experimental observation that the oil nanodroplets' surface potential (& surface charge) remains constant with increasing pH is indeed puzzling and unexpected. The proposed mechanism, in terms of fluctuating polarizability gradients to explain the observed differences in electrophoretic mobilities with pH, appears to be plausible. I recommend this manuscript for publication in Nature Communications, subject to the additional minor comments below.

a) In the newly added text in the manuscript, it is repeatedly stated that the charge conductivity in the bulk aqueous solution increases 2.2x when exchanging Cl⁻ to OH⁻ (see, for instance, page 4, introduction, and page 16, paragraph before Summary). However, the increase in conductivity should be just 2x for NaCl compared to NaOH, and 2.6x if only considering the anions. Including a reference will be valuable. I note, however, that on page 14 (polarization and the Grotthuss mechanism), the authors mentioned a value of x2 instead. A related comment: On page 4, in the last paragraph, it is stated that the polarizability of the bulk solution also increases by x2.2. Is it really so?

Author response: The values of conductivity used come from the “Handbook of Chemistry and Physics”, Ref. 55.

We have systematically checked the manuscript and changed the values as follows:

The ratio between the conductivity of 1 mM NaOH and 1 mM NaCl solutions in water is 2. The polarizability is linearly related to the conductivity, as mentioned in the text.

For the measured mobility ratio, the value is 2.2 ± 0.2 , and we refer to it as this or as 2.2 in the text.

b) In connection to the authors' rebuttal to the second comment of Reviewer#2, they state that the data presented in Figures 1F and 2C, “does not disagree” with previously published by Bakker (2014) and Tian (2020) on flat oil/water interfaces using a similar, yet more established, non-linear optical technique. This is not accurate. The previous studies show, without doubt, the opposite behaviour, clearly indicating that the surface charge increases (becomes more negative) at mildly basic conditions. The authors argue that previous studies did not keep the ionic strength constant. However, at least in the Phys. Rev. Letters 125, 156803 (2020) study, the influence of the ionic strength was explicitly considered in the main text, and particularly in their SI. The authors also appear to suggest potential contaminants could be the cause for the disagreement, but Tian (2020) used submonolayer amounts of purified oil (hexane), representing surface-to-volume ratios similar, if not higher, than those for the nanodroplets, indicating that contamination is unlikely (at least not more than in the nanodroplets). I want to stress that experimental evidence shows that the surface is more negatively charged at flat oil/mildly basic aqueous interfaces, which must be mentioned in the main text. Curved interfaces can behave differently, which has been previously proven by others, including some of the authors in the same manuscript. Thus, it should have no consequences in the manuscript discussions but limit its scope.

Author response: We thank the referee for circling back to this discussion once again. Several points need to be clarified at this point, which were overlooked/incorrectly stated in the previous round of revision:

1. The 2014 study by the Bakker group (J. Chem. Phys. 140, 054711 (2014)) does not disagree with our current findings. This paper showed that the intensity of O-D stretch spectra at the alkane/water interface is higher than the air/water interface. This observation indicated that water at a hydrophobic liquid/water interface is more preferentially ordered than at the air/water interface. Moreover, water molecules at the hydrophobic liquid/water interface form a stronger hydrogen bonding network than at

the air/water interface. These conclusions agree with the findings in our earlier results in Ref.32 [Science 374, 1366-1370 (2021)]. In the current manuscript, we report that the water structure, as measured by the interfacial O-D spectra, is unchanged with an increase in pH. It should be noted that the study from the Bakker group in 2014 did not report any pH dependence of the alkane/water SFG spectra. Therefore, a comparison with the results in this manuscript is not relevant.

2. The second article mentioned by the referee (Phys. Rev. Letters 125, 156803 (2020), and in fact, all concerned studies employing reflection mode SFG measurements) contained no explicit surface charge measurements. The surface charge was deduced from the interfacial O-H stretch spectra, assuming that the surface structure is unchanged upon change in pH. Therefore, the referee's conclusion that: "The previous studies show, without doubt, the opposite behaviour, clearly indicating that the surface charge increases (becomes more negative) at mildly basic conditions", is based on the following (circular) reasoning, which goes as follows:
 - (i) Assume that surface charge solely derives from ions and ionic groups at the interface.
 - (ii) Within that framework, the changes in SFG intensity are assumed solely to originate from changes in surface potential. The intensity changes are then correlated to surface potential and surface charge density using the Guoy Chapman model.
 - (iii) An increasing SFG intensity predicts a higher charge density within the assumed model. Therefore, the surface of basic oil/water interfaces is concluded to be more negatively charged than a neutral one.

As mentioned in our introduction, when not enough variables are independently measured and a restrictive formalism is used, conclusions might be drawn that aren't fully validated.

In the following, we explain why the above type of circular reasoning does not agree with our conclusions.

Differences between Phys. Rev. Letters 125, 156803 (2020) & the present study

The study by Yang et al. (Phys. Rev. Letters 125, 156803 (2020)) showed that the SFG spectrum of water at a hexane/water interface increases in intensity with increased pH. The increase in intensity was modeled using a Guoy-Chapman (GC) model to extract surface charge densities. Using the GC model excludes any charge transfer explanation, as it permits only ionic species as charge carriers embedded in a uniform dielectric.

A second assumption behind this analysis is that the surface susceptibility, $\chi_s^{(2)}$, and thus also the surface structure, remains unchanged when the bulk pH of the solution is changed. However, Fig. S4 of Phys. Rev. Letters 125, 156803 (2020), reprinted here as Fig. R1, showed that hexane alkyl chains drastically reorient as a function of pH. At pH 5.6, the alkyl chains are

nearly perpendicular to the interface, and become nearly parallel to the interface at pH 10. This means that $\chi_s^{(2)}$ is pH-dependent.

Fig. R1. The reorientation of hexane at the air/water interface as a function of pH. Figure reproduced from *Phys. Rev. Letters* 125, 156803 (2020).

Furthermore, this type of reorientation can significantly alter the interfacial water ordering. Both Bakker (2014) and Tian (2020) have shown that the SFG intensity at the hydrophobic liquid/water interface is stronger than the air/water interface due to the water ordering by hydrophobic groups, and it also depends on how these groups are oriented. At the planar hexane/water interface, the alkyl chains adopt an increasingly parallel orientation with increased pH, creating more C-H-water contact points. This increases the total ordered water at the interface and hence the O-H stretch intensity.

In the nanoscale system studied in this work, the alkyl chains at the oil droplet surface are parallel to the surface, and their orientation is not affected by the pH of the surrounding solution as evident from the C-H spectra in Fig. 2 (intensity and spectral shape are pH-independent). The water ordering at the interface as measured by the SFS spectral shape and intensity in the O-D spectral region is unchanged.

With the above re-interpretation of the results by Tian (PRL, 2020), there is no contradiction with the manuscript. The difference arises from the assumptions made by Tian et al.

We have added the discussion about this in the supplementary information section S1.

Finally, we note that Tian and co-workers perform instructive AIMD simulations that appear to suggest some tendency for the OH⁻ to bind to the hexane-water interface. The surface affinity of OH⁻ to hydrophobic interfaces has been a topic of a lot of controversy with different calculations (with varying levels of underlying potential and simulation timescales), presenting disparate results [See for example *J. Am. Chem. Soc.* 2015, 137, 39, 12610–12616; *Chemical Physics Letters* 481 (2009) 2–8; *J. Phys. Chem. B* 2014, 118, 28, 8364–8372]. Nonetheless, most simulations either present a very weak binding of OH⁻ to the surface of water ($k_B T$) or in fact show that OH⁻ ions are repelled from the interface. Our view of the matter is that this surface affinity is too small to account for the magnitude of the ζ -potentials measured. However, this is a topic that clearly deserves attention with more sophisticated calculations.

Regarding the surface-to-volume ratio, the paper by Tian et al., PRL 2020 uses a planar surface whose area is not given, but we estimate that it has to be at least as large as the optical beams. Then, the relevant volume to be considered is the volume of liquid in the flask for the

hexane, and the amount of water in the water cell. It is *not* the volume of the layer on the surface as appears to be the assumption of the referee. The metric of relevance in terms of impurity discussions is ‘how much impurities are there in the total sample volume’ and ‘how much area is there to adsorb upon,’ in addition to ‘how likely is it that this impurity adsorbs.’ We have clarified this in the supplementary information section S1.

c) Authors' rebuttal to the third comment from Reviewer#2 (a lack of a clearly discernible peak linked to the OH- in the sum frequency scattering is not proof of its absence at the interface because, to my knowledge, it has never been reported). The authors provide a reference for CaF₂-water surfaces (Khalib et al., though first reported by Becraft and Richmond, Langmuir 2001, 17, 25, 7721). However, in those studies, the ‘‘narrow peak at 3645 cm⁻¹’’ originates from the CaOH, not the OH- anion. Since the authors cannot find a suitable reference, my fears appear justified: the lack of a discernible peak in the sum frequency scattering spectra is not proof of its absence at the interface. The concerned text should be edited accordingly.

The initial reviewer question was:

‘‘To my knowledge, there are no reports of a relatively narrow hydroxide anion stretch in the sum frequency literature, even in systems where the OH- is certain to be present at the surface. Maybe the authors can provide a reference?’’

We answered with

‘‘Author response:

The IR and Raman cross-sections and peak shapes are in fact equally large, this can be found e.g. in Ref. [Hermannson et al, Chemical Physics Letters 514 (2011) 1–15], and was also reproduced experimentally in our lab. There are SFG spectra of CaF₂ – water surfaces whereby the F⁻ ions are replaced by OH⁻ ions at high pH (Khatib, R. et al. Water orientation and hydrogen-bond structure at the fluorite/water interface. Sci. Rep. **6**, 24287; doi: 10.1038/srep24287 (2016).). This replacement gives rise to a narrow peak at ~ 3645 cm⁻¹, precisely where OH⁻ is indeed expected. On these grounds, it would also have to appear at the oil-water interface if it is there.’’

To the current referee's remark ‘‘However, in those studies, the ‘‘narrow peak at 3645 cm⁻¹’’ originates from the CaOH, not the OH- anion.’’ we add the following:

The vibrational mode at ~3645 cm⁻¹ is present in both the IR/Raman spectrum of Ca(OH)₂ and Mg(OH)₂ crystals, and is unambiguously assigned to the A_{1g}(OH) stretch vibration. For Ca(OD)₂ the same vibration occurs at 2688 cm⁻¹ (R. A. Buchanan, H. H. Caspers, and J. Murphy, Lattice Vibration Spectra of Mg(OH)₂ and Ca(OH)₂, Applied Optics Vol. 2, Issue 11, pp. 1147-1150 (1963)). Molecular crystals retain the vibrational modes of the individual molecular ions, especially when they consist of highly localized modes > 1000 cm⁻¹, such as the O-H symmetric stretch modes.

Therefore, we can make the following statements:

- 1- IR and Raman spectra of strongly basic solutions both contain a sharp peak feature at ~ 3645 / 2688 cm⁻¹ for OD⁻/OH⁻. Such spectra were also recorded in our lab and can be provided upon request. They are also in the literature as reviewed by Hermannson as per our original answer.
- 2- SFG involves a combination of IR and Raman processes, and so it should provide a clear intensity at this wavenumber if this particular species is present in an anisotropic orientational distribution.
- 3- Ca(OH)₂ / Mg(OH)₂ crystals contain the same (O-H⁻ A_{1g}) peak at nearly the same frequency (3644 / 3650 cm⁻¹).

- 4- $\text{Ca}(\text{OD})_2$ contains a similar peak that is shifted by the expected amount due to isotope substitution (2688 cm^{-1}).
- 5- This peak is assigned to the symmetric O-H stretch vibration (O-H $^-$ A_{1g} mode)
- 6- The reflection mode SFG spectrum of a CaF_2 crystal in a very basic solution contains a sharp peak at $\sim 3645 \text{ cm}^{-1}$.
- 7- $\text{OH}^- / \text{OD}^-$ can be observed at a surface by reflection mode SFG.
- 8- The absence of such a peak at a different surface provides a form of evidence that it is possibly not present.

In addition to the text currently in the manuscript:

“However, neither hydroxide nor surface-active carboxylate impurities were detected on the surface in vibrational sum frequency scattering measurements³¹⁻³³, even though the necessary concentration at $\text{pH} > 9$ is well within the detection limit of this surface sensitive method (See supporting information section S1 for more details on impurities and how their role is eliminated in our findings).”

We have added to the SI:

“Reflection SFG studies on the hexane-water interface as a function of pH

Two recent studies have performed femtosecond reflection mode SFG experiments on the hexane/water interface, whereby the hexane was dosed from the gas phase to form films with sub monolayer to monolayer coverages^{25, 26}. It was shown by both Ref.²⁵ and Ref.²⁶ that the SFG intensity at the hydrophobic liquid/water interface is stronger than the air/water interface due to the water ordering by hydrophobic groups. The 2020 study by Yang et al.²⁶ additionally showed that the SFG spectrum of water at a hexane/water interface increases in intensity with increasing pH, and constant ionic strength. Although there was no direct surface potential and streaming potential measurement, the increase in intensity was attributed to the interfacial presence of ionic charge carriers, and modelled using a Guoy-Chapman model to extract surface charge densities. It was also assumed that the surface susceptibility was pH independent (meaning that $\chi_s^{(2)}$ is constant). Ref.²⁶ also showed that hexane chains drastically reorient as a function of pH. At pH 5.6, the alkyl chains are nearly perpendicular to the interface, and they become more parallel to the interface at pH 10. Such a reorientation has two consequences: it likely changes the value of $\chi_s^{(2)}$ as a function of pH, and it can significantly alter the interfacial water ordering. The SFG data was further analysed to obtain an adsorption free energy, which had the value of 37 kJ/mol ($\sim 14 \text{ kT}$), which should lead to a density of $\sim 2 \text{ OH}^- \text{ ions / nm}^2$. With such densities, considering the sensitivity of SFG, one might certainly expect a spectral signature of interfacial OH^- . OH^- has a vibrational mode at 3645 cm^{-1} , which is Raman and IR active, and can be detected with IR/Raman spectroscopy in crystals of $\text{Ca}(\text{OH})_2$, $\text{Mg}(\text{OH})_2$, as well as on the CaF_2 /water interface that is immersed in basic aqueous solution³⁴⁻³⁶. Since such a signature is not present at high pH, despite the (experimentally) expected high surface propensity, one might propose a re-evaluation of the experimental evidence with less stringent assumptions. Compared to the present work, the change in alkane orientation is not observed (Fig. 2). Both works are therefore not necessarily in disagreement.”

REVIEWER COMMENTS

Reviewer #2 (Remarks to the Author):

I am puzzled by the authors' response to what I thought were otherwise constructive comments.

a) In the third paragraph of the introduction (lines 92 to 95), the authors state:

"However, neither hydroxide nor surface-active carboxylate impurities were detected on the surface in vibrational sum frequency scattering measurements, even though the necessary concentration at $\text{pH} > 9$ is well within the detection limit of this surface sensitive method."

As mentioned previously, spectral features from the OH ANION have not been reported in the Sum Frequency literature, even in concentrated (above $\text{pH} 14$) basic solutions (for instance, Richmond et al., Allen et al., Bonn et al., etc). Thus, the text is misleading and factually incorrect, as the hydroxide anion is not within the "detection limit" of the sum frequency scattering measurements at pH just above 9 as implied in the text. Please make the necessary changes.

As a side note, the IR spectra of concentrated basic solutions do not show a sharp feature linked to the OH-, as the authors mentioned in their rebuttal (only the Raman does). In IR, it is rather a broad and "continuum" response similar to what is observed for the proton. I invite the authors to consult the actual experimental references (among many others available in the literature) cited in the review by Hermansson et al. (Chem. Phys. Letters 514, 2011, 1)

Additionally, though the OH stretch in mineral and oxide surfaces (CaOH, MgOH, MnOH, SiOH, AlOH, etc), is readily detectable with sum frequency generation, it does not imply that the hydrated anion will have the same sum frequency cross-section, peak position, and bandwidth (in fact they all vary significantly between the different hydroxylated surfaces).

b) When discussing the data in Figure 2C (one of the experimental centrepieces of the manuscript), I insist you mention somewhere in the text that at flat surfaces, the spectrum of water at the hexane/water interface, in contrast to the droplets, increases in intensity with increasing pH (Phys. Rev. Letters 125, 156803 (2020)). You can then also refer to the text you added to the SI, which, at the moment, is disconnected from the manuscript. It is curious that the authors don't cite the paper from Yang et al. (Phys. Rev. Letters 125, 156803 (2020)) anywhere in the main text, although they investigate the same system on flat surfaces using a sister technique, but reach an opposite conclusion. You may well disagree with the interpretation of the results presented by Yang et al., but the data is as compelling as yours.

Author response to reviewer comments:

We thank the referee for examining our manuscript in great detail. In the following, we reproduce the changes made to the manuscript and SI in response to the reviewer's comments. The reviewer's comments are in blue and our responses are in black.

Reviewer #2 (Remarks to the Author):

I am puzzled by the authors' response to what I thought were otherwise constructive comments.

a) In the third paragraph of the introduction (lines 92 to 95), the authors state:

"However, neither hydroxide nor surface-active carboxylate impurities were detected on the surface in vibrational sum frequency scattering measurements, even though the necessary concentration at pH > 9 is well within the detection limit of this surface sensitive method."

As mentioned previously, spectral features from the OH ANION have not been reported in the Sum Frequency literature, even in concentrated (above pH 14) basic solutions (for instance, Richmond et al., Allen et al., Bonn et al., etc). Thus, the text is misleading and factually incorrect, as the hydroxide anion is not within the "detection limit" of the sum frequency scattering measurements at pH just above 9 as implied in the text. Please make the necessary changes.

As a side note, the IR spectra of concentrated basic solutions do not show a sharp feature linked to the OH-, as the authors mentioned in their rebuttal (only the Raman does). In IR, it is rather a broad and "continuum" response similar to what is observed for the proton. I invite the authors to consult the actual experimental references (among many others available in the literature) cited in the review by Hermansson et al. (Chem. Phys. Letters 514, 2011, 1)

Additionally, though the OH stretch in mineral and oxide surfaces (CaOH, MgOH, MnOH, SiOH, AlOH, etc), is readily detectable with sum frequency generation, it does not imply that the hydrated anion will have the same sum frequency cross-section, peak position, and bandwidth (in fact they all vary significantly between the different hydroxylated surfaces).

Author response:

To clarify the confusion regarding OH- anions and detection limit, we have modified the text as:

"However, neither a spectral signature corresponding to hydroxide nor surface-active carboxylate impurities were detected on the surface in vibrational sum frequency scattering measurements³¹⁻³³, even though molecular groups with much lower cross-sections and in bulk concentrations as low as ~10 μ M, were detected using this surface-sensitive method³⁴. (See supporting information section S1 for more details on impurities and how their role is eliminated in our findings)."

The SI paragraph discussing planar hexane/water interface results has been changed:

Reflection SFG studies on the hexane-water interface as a function of pH

Two recent studies have performed femtosecond reflection mode SFG experiments on the hexane/water interface, whereby the hexane was dosed from the gas phase to form films with sub monolayer to monolayer coverages^{25, 26}. It was shown by both Ref.²⁵ and Ref.²⁶ that the SFG intensity at the hydrophobic liquid/water interface is stronger than the air/water interface due to the water ordering by hydrophobic groups. The 2020 study by Yang et al.²⁶ additionally showed that the SFG spectrum of water at a hexane/water interface increases in intensity (but not in shape) with increasing pH at constant ionic strength. Although there was no direct surface potential and streaming potential measurement, the increase in intensity was attributed to the interfacial presence of ionic charge carriers, and modelled using a Gouy-Chapman model to extract surface charge densities. It was also assumed that the surface

susceptibility was pH-independent (meaning that $\chi_s^{(2)}$ is constant). Interestingly, Ref.²⁶ also showed that hexane chains drastically reorient as a function of pH. At pH 5.6, the alkyl chains are nearly perpendicular to the interface, and they become more parallel to the interface at pH 10. Such a reorientation has two consequences: it likely changes the value of $\chi_s^{(2)}$ as a function of pH, and it can significantly alter the interfacial water ordering. The SFG data was further analysed to obtain an adsorption free energy, which had the value of 37 kJ/mol (~ 14 kT), which should lead to a surface density of ~ 2 OH⁻ ions / nm². With such densities, considering the sensitivity of SFG, one might expect a spectral signature of interfacial OH⁻. SFG reports on the product of IR and Raman cross-sections. OH⁻ has a vibrational mode at 3645 cm⁻¹, which is Raman active (narrow peak on the blue spectral side with a strong absorption cross section) and IR active (shoulder on the blue spectral side having a weaker absorption cross section than in Raman spectroscopy), yet it hasn't been detected in the SF spectra of hydrophobic/water interfaces. IR and Raman spectroscopy in crystals of Ca(OH)₂, Mg(OH)₂ show the hydroxide O-H stretch mode. SFG from the CaF₂/water interface immersed in basic aqueous solution has also shown a hydroxide feature³⁴⁻³⁶. Although this could be either surface adsorbed hydroxide or hydroxide within the surface lattice, it suggests that surface hydroxide species are SFG active and detectable when present. Therefore, it is curious that a molecular group that is SFG active with a surface adsorption free energy of 14kT does not produce a sum frequency mode corresponding to the adsorbed OH⁻ anion at hydrophobic/water interfaces.

On the other hand, the absence of a hydroxide peak might not provide conclusive evidence for the absence of OH⁻ at the surface. The lack of spectral signature can also be due to other causes, such as the lack of a specific orientation combined with an unexpectedly low cross-section. Therefore, only spectral measurements in combination with independent surface potential measurements or other direct probes of OH⁻ can conclusively prove the presence/absence of OH⁻ ions. At the planar hexane/water interface, such independent measurements are yet to be made. Therefore, the increase in the O-H stretch signal at the planar interface can only be viewed as an increased ordering of hydrogen-bonded water at the interface, which might have its origins in the reorientation of alkyl chains rather than OH⁻ absorption. In the present work, however, no change in the alkane orientation is observed as a function of pH (Fig. 2D). Both studies are therefore not necessarily in disagreement.

b) When discussing the data in Figure 2C (one of the experimental centrepieces of the manuscript), I insist you mention somewhere in the text that at flat surfaces, the spectrum of water at the hexane/water interface, in contrast to the droplets, increases in intensity with increasing pH (Phys. Rev. Letters 125, 156803 (2020)). You can then also refer to the text you added to the SI, which, at the moment, is disconnected from the manuscript. It is curious that the authors don't cite the paper from Yang et al. (Phys. Rev. Letters 125, 156803 (2020)) anywhere in the main text, although they investigate the same system on flat surfaces using a sister technique, but reach an opposite conclusion. You may well disagree with the interpretation of the results presented by Yang et al., but the data is as compelling as yours.

Author response:

On page 10, we added:

"Yang et al.⁴³ observed that the sum frequency O-H stretch spectra at a planar pristine hexane/water interface increased in overall intensity with increased pH. The differences between this study and our present results might originate from either the difference in the model system (nanospheres vs extended planar interface) or the response of the hexane, as detailed in the SI section S1. Further studies might be required to reconcile the differences."

REVIEWERS' COMMENTS

Reviewer #2 (Remarks to the Author):

I am satisfied with the changes made by the authors.